# Unified (Semi) Unbalanced and Classic Optimal Transport with Equivalent Transformation Mechanism and KKT-Multiplier Regularization

## Abstract

Semi-Unbalanced Optimal Transport (SemiUOT) shows great promise in matching two probability measures by relaxing one of the marginal constraints. Previous solvers often incorporate an entropy regularization term, which can result in inaccurate matching solutions. To address this issue, we focus on determining the marginal probability distribution of SemiUOT with KL divergence using the proposed Equivalent Transformation Mechanism (ETM) approach. Furthermore, we extend the ETM-based method into exploiting the marginal probability distribution of Unbalanced Optimal Transport (UOT) with KL divergence for validating its generalization. Once the marginal probabilities of UOT/SemiUOT are determined, they can be transformed into a classical Optimal Transport (OT) problem. Moreover, we propose a KKT-Multiplier regularization term combined with Multiplier Regularized Optimal Transport (MROT) to achieve more accurate matching results. We conduct several numerical experiments to demonstrate the effectiveness of our proposed methods in addressing UOT/SemiUOT problems.

## 1. Introduction

Optimal Transport (OT) technique is a powerful tool for discerning and comparing distinct probability distributions. Nowadays, OT has multiple successful applications in traditional machine learning (Frogner et al., 2015; Feydy et al., 2019; Zhuang et al., 2022; Chuang et al., 2023; Riaz et al., 2023), unsupervised clustering (Asano et al., 2019; Caron et al., 2020), domain adaptation (Damodaran et al., 2018; Courty et al., 2017; Redko et al., 2019), diffusion (Khrulkov et al., 2023; Lipman et al., 2023), generative modeling (Korotin et al., 2023; Onken et al., 2021; Tong et al., 2023)

[1]Anonymous Institution, Anonymous City, Anonymous Region, Anonymous Country. Correspondence to: Anonymous Author <anon.email@domain.com>.

Preliminary work. Under review by the International Conference on Machine Learning (ICML). Do not distribute.

and many others. Nevertheless, directly solving OT distances could have relatively high computation cost with around super-cubic time. Although one can adopt entropy-based Sinkhorn algorithm (Cuturi, 2013) for solving OT efficiently, it still suffers from the dilemma of dense and inaccurate solutions (Liu et al., 2023; Lorenz et al., 2021; Dessein et al., 2018). Moreover, classical OT strictly assume that the masses on both source and target domains should be equal. It further hurdles the generalization of OT when the data samples inherit noise or outliers.

Recently, Unbalanced Optimal Transport (UOT) (Benamou, 2003; Chizat, 2017; Séjourné et al., 2023; Scetbon et al., 2023; Séjourné et al., 2022b) and Semi-Unbalanced Optimal Transport (SemiUOT) (Le et al., 2021) have become more attractive in adapting outliers since it allows relaxing marginal constraints for transportation results. UOT adopts several divergences such as Kullback-Leiber (KL) divergence (Pham et al., 2020), $\ell_1$ norm (Caffarelli & McCann, 2010) and $\ell_2$ norm (Blondel et al., 2018) for the relaxation on OT mass equality constraints by adjusting the corresponding coefficients $\tau$. Meanwhile, KL divergence is the most commonly-used in UOT formulation in real practice (Séjourné et al., 2022a). UOT also provides great applications in transfer learning (Tran et al., 2023; Mukherjee et al., 2021; Pariset et al., 2023), computer vision (Bonneel & Coeurjolly, 2019; De Plaen et al., 2023; Choi et al., 2023; Neklyudov et al., 2023; Chang et al., 2022; Ma et al., 2021; Zhan et al., 2021), structure data exploration (Sato et al., 2020), natural language processing (Arase et al., 2023) and many areas. Previous solvers always involves extra regularization terms including entropy regularization term and proximal point term (Fatras et al., 2021) for tackling UOT problem. While adding additional entropy terms will lead to dense and inaccurate matching solutions. Latest, (Chapel et al., 2021) and (Nguyen et al., 2023) further reconsider solving UOT problem with majority maximization algorithm without the requirements of regularization terms. However, these methods are sensitive to the choice of $\tau$, i.e., providing sparse and accurate solutions when $\tau$ is small, but unsatisfying solutions when $\tau$ is much larger. Therefore, it is quite challenging to efficiently exploit accurate solutions for both UOT and SemiUOT problems.

In this paper, we propose a new method, i.e., *equivalent transformation mechanism* (ETM), which relieves the need for extra regularization terms in solving SemiUOT and UOT with KL divergence. Specifically, ETM first finds the marginal probability distributions for SemiUOT and UOT problems based on Karush-Kuhn-Tucker (KKT) conditions and their dual forms. We can further observe that the essence of SemiUOT and UOT lies in correspondingly adjusting the initial weights of different data samples. This provides us with new insights for understanding SemiUOT and UOT problems, i.e., we can transform SemiUOT and UOT problems into classic optimal transport problems based on initial marginal weights. Though we can exactly solve the marginal distributions via conventional iterative methods, e.g., L-BFGS, we further propose ETM-Approx to achieve the approximate results efficiently and convergently. Moreover, ETM-Refine resolves exact solutions via quasi-Newton optimization where the start points are these approximate solutions. Compared with original ETM, ETM-Apporx and ETM-Refine obtain accurate solutions while competitively balancing computational cost. Beyond solving the marginal distribution, we also discover that the KKT multipliers provide valuable guidance for addressing the OT problem, which is transformed from the SemiUOT and UOT problems with marginal weights. Therefore we further proposed *Multiplier Regularized Optimal Transport* (MROT) for achieving more sparse and accurate OT matching solutions. We summarize our contributions: (1) To our best knowledge, we first propose both exact and approximate solutions for ETM on two problems, i.e., SemiUOT and UOT. After optimizing these problems, one can obtain the sample marginal probabilities and transfer SemiUOT/UOT into standard optimal transport problems. (2) We first innovatively propose multiplier constraint terms to establish MGOT for achieving more accurate results. (3) We conduct extensive experiments on both synthetic and real-world datasets to demonstrate the performance of proposed ETM.

## 2. Preliminary

We first provide a brief preliminary definition of OT, UOT and SemiUOT. Let us consider two sets of data samples $X \in \mathbb{R}^{M \times D}$ and $Z \in \mathbb{R}^{N \times D}$ in source and target domains, where $M$, $N$ denote the number of samples and $D$ denotes the data dimension. Each data samples has corresponding prior-given mass weights $a \in \mathbb{R}^{M \times 1}$ and $b \in \mathbb{R}^{N \times 1}$. Meanwhile the total masses of these data samples are equal as $a^\top \mathbf{1}_M = b^\top \mathbf{1}_N$. The classical OT problem was defined by (Kantorovich, 1942) with a linear problem to measure the minimum transportation cost among data sample $X$ and $Z$:

$$\min_{\pi_{ij} \geq 0} J_{\mathrm{OT}} = \langle C, \pi \rangle \quad \text{s.t. } \pi \mathbf{1}_N = a, \quad \pi^\top \mathbf{1}_M = b,$$

where $C \in \mathbb{R}^{M \times N}$ denotes the pairwise distance matrix. Meanwhile $\pi \in \mathbb{R}^{M \times N}$ denotes the coupling matching matrix among the data samples $X$ and $Z$. One can directly

solve $J_{\mathrm{OT}}$ via utilizing network-flow algorithm (Kenning-ton & Helgason, 1980; Ahuja et al., 1988). To consider more general cases (e.g., filtering out the noise or outliers), one can relax two marginal constraints, i.e., $\pi \mathbf{1}_N \neq a$ and $\pi^\top \mathbf{1}_M \neq b$, to achieve unbalanced optimal transport problem (Pham et al., 2020):

$$\min_{\pi_{ij} \geq 0} J_{\mathrm{UOT}} = \langle C, \pi \rangle + \tau_a \mathrm{KL}\left(\pi \mathbf{1}_N \| a\right) + \tau_b \mathrm{KL}(\pi^\top \mathbf{1}_M \| b),$$

where $\mathrm{KL}\left(\cdot\right)$ denotes Kullback-Leiber (KL) divergence which has been widely used in dealing with UOT. $\tau_a$ and $\tau_b$ denote the balanced hyper parameters between the minimizing cost and marginal relaxation. Note that when $\tau_a, \tau_b \rightarrow +\infty$ and $a^\top \mathbf{1}_M = b^\top \mathbf{1}_N$, UOT problem will turn into classical OT. Moreover, we can add one marginal constraints to formulate SemiUOT. For instance, we relax the constraint $\pi \mathbf{1}_N \neq a$ and keep the constraint $\pi^\top \mathbf{1}_M = b$:

$$\min_{\pi_{ij} \geq 0} J_{\mathrm{SemiUOT}} = \langle C, \pi \rangle + \tau \mathrm{KL}\left(\pi \mathbf{1}_N \| a\right) \quad \text{s.t. } \pi^\top \mathbf{1}_M = b.$$

Previous researches always add entropy regularization term for solving OT, UOT and SemiUOT. Although entropy regularization term can enhance the scalability of solving $\pi^*$, it still suffers from the dense and inaccurate solution dilemma. In the following, we will first investigate the problem of UOT/SemiUOT from the perspective of marginal probability distribution, in order to find out the accurate solution of $\pi^*$ for OT, UOT and SemiUOT.

## 3. Methodology

In this section, we will provide the calculation details on finding the solutions for commonly-existed UOT and SemiUOT. Previous methods (Pham et al., 2020; Chizat et al., 2018) always directly adopted entropy-based regularization term into tackling UOT and SemiUOT problem. Although such approaches can provide fast computation speed, it will lead to relatively ambiguous and dense solution which does not match most of situations in real practices (Li et al., 2023; Scetbon et al., 2021). Latest, (Chapel et al., 2021) adopted majorization-minimization algorithm or regularization path for solving UOT/SemiUOT problem. However, majorization-minimization algorithm is sensitive to the choice of $\tau$, and still causes inaccurate and dense solutions when $\tau \rightarrow +\infty$. Worse still, regularization path could involve heavy matrix computation on inversion, requiring complicated optimization procedure. To solve the above problem, we change the perspective of solving the UOT/SemiUOT problem, i.e., originally exploiting the marginal probability of UOT/SemiUOT via our proposed *Equivalent Transformation Mechanism* (ETM) approach. In this way, we can obtain some interesting insights on understanding the intrinsic characteristics of UOT/SemiUOT. Moreover, we further propose *KKT-Multiplier Regularization* with *Multiplier Regularized Optimal Transpor* (MROT) with theorems and corollaries to achieve more accurate matching solution on SemiUOT and UOT respectively.

### 3.1. Equivalent Transformation Mechanism

In this section, we will first introduce the proposed equivalent transformation mechanism approach. Specifically, we propose an ETM-based method to determine the marginal probabilities of source data samples in SemiUOT, accompanied by detailed illustrations. We then extend the ETM-based method to address the more complex UOT problem.

**Equivalent Transformation Mechanism for SemiUOT.** To start with, we first exploit the marginal probability distributions for SemiUOT via the proposed ETM method. Specifically, ETM includes three different approaches, i.e., ETM-Exact, ETM-Approx and ETM-Refine. By utilizing the methods above, one can transform SemiUOT into classic optimal transport problem. In this section, we will introduce the deduction and optimization details for the proposed ETM-based method on SemiUOT.

**Proposition 1.** (Principles of Equivalent Transformation Mechanism for SemiUOT) *Given SemiUOT with KL-Divergence $J_{\mathrm{SemiUOT}}$, one can obtain its Fenchel-Lagrange multipliers form as:*

$$\min_{\boldsymbol{f},\boldsymbol{g},\zeta} \left[ \tau \sum_{i=1}^{M} a_i e^{-\frac{f_i+\zeta}{\tau}} - \sum_{j=1}^{N} b_j(g_j - \zeta) \right] \quad (1)$$

$$s.t. \ f_i + g_j + s_{ij} = C_{ij}, \quad s_{ij} \geq 0.$$

*where $\boldsymbol{f}$, $\boldsymbol{g}$, $\boldsymbol{s}$ and $\zeta$ denotes Lagrange multipliers. Moreover, SemiUOT problem can be transformed into classic optimal transport as follows:*

$$\min_{\boldsymbol{\pi} \geq 0} \mathcal{J}_{\mathrm{P}} = \langle \boldsymbol{C}, \boldsymbol{\pi} \rangle \quad s.t. \begin{cases} \boldsymbol{\pi}\mathbf{1}_N = \boldsymbol{a} \odot \exp\left(-\dfrac{\boldsymbol{f}^* + \zeta^*}{\tau}\right) = \boldsymbol{\alpha} \\ \boldsymbol{\pi}^\top \mathbf{1}_M = \boldsymbol{b} \end{cases}$$
$$(2)$$

*Note that when $\tau \to \infty$, the source marginal probability can be determined as $\boldsymbol{\pi}\mathbf{1}_N = \omega\boldsymbol{a}$ where $\omega = \langle \boldsymbol{b}, \mathbf{1}_N \rangle / \langle \boldsymbol{a}, \mathbf{1}_M \rangle$.*

The proof of Proposition 1 can be found in Appendix A. We can observe that SemiUOT is set to assign different weights on data samples. To further simplify the calculation by reducing variable $\boldsymbol{g}$, we set $g_j = \inf_{k \in [M]} (C_{kj} - f_k)$ according to the $c$-transform theorem (Villani et al., 2009). Therefore, we only need to optimize $\boldsymbol{f}$ and $\zeta$ without additional constraints as follows:

$$\min_{\boldsymbol{f},\zeta} L_{\mathrm{P}} = \tau \sum_{i=1}^{M} a_i e^{-\frac{f_i+\zeta}{\tau}} - \sum_{j=1}^{N} \left[ \inf_{k \in [M]} [C_{kj} - f_k] - \zeta \right] b_j, \quad (3)$$

We refer to $L_{\mathrm{P}}$ as the newly proposed *Exact SemiUOT Equation*. Specifically, we initialize $\zeta = 0$ for the optimization. We first fix $\zeta$ then adopting L-BFGS method (Berahas et al., 2016; Virtanen et al., 2020) to reach optimal results of $\boldsymbol{f}^\ell$ and $g_j^\ell = \inf_{k \in [M]}(C_{kj} - f_k^\ell)$ at the $\ell$-th iteration. Then we optimize $\zeta = \tau[\log(\sum_{i=1}^{M} a_i \exp(-f_i^\ell/\tau)) - \log(\sum_{j=1}^{N} b_j)]$ which is obtained by considering $\nabla_\zeta L_{\mathrm{P}} = 0$ and it guarantees $\sum_{i=1}^{M} a_i e^{-(f_i+\zeta)/\tau} = \sum_{j=1}^{N} b_j$. We it-

eratively update $L_{\mathrm{U}}$ to reach the optimal solution on $\zeta^*$, $\boldsymbol{f}^*$ and $\boldsymbol{g}^*$. Here we refer to the entire optimization procedure as the ETM-Exact approach for addressing the Exact Semi-UOT Equation. Although $L_{\mathrm{P}}$ is convex and has unique solutions, the presence of $\inf(\cdot)$ renders it a non-smooth function, leading to inefficient optimization (An et al., 2022). To further accelerate the optimization process, we consider to make a smooth approximation on replacing $\inf(\cdot)$ as $\inf_{k \in [M]}[C_{kj} - f_k] \approx -\epsilon \log[\sum_{k=1}^{M} e^{\frac{f_k - C_{kj}}{\epsilon}}]$. Note that $\epsilon > 0$ denotes the balanced hyper parameters among the accuracy and function smoothness. Smaller $\epsilon$ (e.g., $\epsilon$ approaches to 0) could lead to more accurate while less smooth solutions. Then we can obtain the proposed *Approximate SemiUOT Equation* as $\widehat{L}_{\mathrm{P}}$ by replacing $\inf(\cdot)$ with the smoothness term for $\widehat{f}$ as below:

$$\min_{\widehat{\boldsymbol{f}},\zeta} \widehat{L}_{\mathrm{P}} = \tau \sum_{i=1}^{M} a_i e^{-\frac{\widehat{f}_i+\zeta}{\tau}} + \sum_{j=1}^{N} b_j \left[ \log\left[ \sum_{k=1}^{M} e^{\frac{\widehat{f}_k - C_{kj}}{\epsilon}} \right]^\epsilon + \zeta \right]$$
$$(4)$$

**Proposition 2.** (Calculation for Approximate SemiUOT Equation) *Given Approximate SemiUOT equation $\widehat{L}_{\mathrm{P}}$, it can be optimized via Equivalent Transformation Mechanism with Approximation (ETM-Approx). That is, ETM-Approx aims to solve the following equation for each $\widehat{f}_s$:*

$$\frac{\partial \widehat{L}_{\mathrm{P}}}{\partial \widehat{f}_s} = -a_s e^{-\frac{\widehat{f}_s+\zeta}{\tau}} + e^{\frac{\widehat{f}_s}{\epsilon}} \sum_{j=1}^{N} \left[ \frac{b_j \exp\left(-\frac{C_{sj}}{\epsilon}\right)}{\sum_{k=1}^{M} \exp\left(\frac{\widehat{f}_k - C_{kj}}{\epsilon}\right)} \right] = 0. \quad (5)$$

*Specifically, we can adopt fixed-point iteration method for solving Eq.(5) at the $\ell$-th iteration as follows:*

$$\begin{cases} \widehat{f}_1^{\ell+1} = \nu \left[ \log\left( a_1 e^{-\frac{\zeta}{\tau}} \right) - \log\left[ \sum_{j=1}^{N} \left( \frac{b_j e^{-C_{1j}/\epsilon}}{\mathscr{W}_{\epsilon,j}(\widehat{\boldsymbol{f}}^\ell)} \right) \right] \right] \\ \vdots \\ \widehat{f}_M^{\ell+1} = \nu \left[ \log\left( a_M e^{-\frac{\zeta}{\tau}} \right) - \log\left[ \sum_{j=1}^{N} \left( \frac{b_j e^{-C_{Mj}/\epsilon}}{\mathscr{W}_{\epsilon,j}(\widehat{\boldsymbol{f}}^\ell)} \right) \right] \right], \end{cases} \quad (6)$$

*where $\nu = \tau\epsilon/(\tau + \epsilon)$ for simplification and $\mathscr{W}_{\epsilon,j}(\widehat{\boldsymbol{f}}^\ell)$ denotes the corresponding calculation as shown below:*

$$\mathscr{W}_{\epsilon,j}(\widehat{\boldsymbol{f}}^\ell) = \sum_{k=1}^{M} \exp\left( \frac{\widehat{f}_k^\ell - C_{kj}}{\epsilon} \right). \quad (7)$$

*The proposed procedure can be convergence with theoretical guarantee after $\mathcal{T}$-th inner iteration. Finally, updating variable $\zeta$ by further considering $\nabla_\zeta \widehat{L}_{\mathrm{P}} = 0$ via $\zeta = \tau[\log(\sum_{i=1}^{M} a_i \exp(-\widehat{f}_i^*/\tau)) - \log(\sum_{j=1}^{N} b_j)]$. One can achieve the optimal solution on $\widehat{\boldsymbol{f}}$ and $\zeta$ accordingly.*

The proof of Proposition 2 can be found in Appendix B. Generally, Proposition 2 outlines the optimization procedure using the newly proposed ETM-Approx approach for addressing the Approximate Semi-UOT Equation. We can observe that the ETM-Approx approach is easy to compute

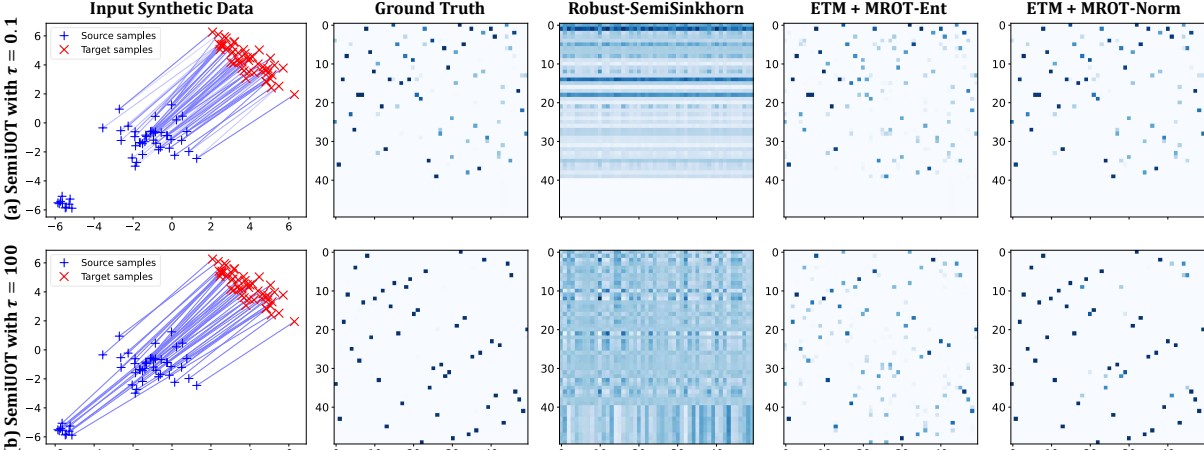

Figure 1. The SemiUOT matching solutions on $\boldsymbol{\pi}^*$ when $\tau = 0.1$ or $\tau = 100$ among the Robust-SemiSinkhorn (Le et al., 2021) and our proposed ETM + MROT-Ent, ETM + MROT-Norm with $\eta_G = 10^2$ and $\epsilon = 10^{-2}$. We set $\eta_{\text{Reg}} = 0.1$ for entropy or $L_2$-norm regularization term. Our proposed method can avoid ambiguous matching solution and achieve more accurate results.

and implement, while avoiding complex calculations (such as finding the step size and estimating the Hessian matrix) and not requiring a large amount of storage space against previous methods. Therefore, the ETM-Approx approach is an efficient method for determining the optimal result of $\widehat{\boldsymbol{f}}^*$ and $\widehat{g}_j^* = -\epsilon \log[\sum_{k=1}^M \exp((\widehat{f}_k^* - C_{kj})/\epsilon)]$, transforming SemiUOT into a classical optimal transport problem.

Moreover, we can finally figure out the exact optimal solution $\boldsymbol{f}^*$ via the approximate optimal solution $\widehat{\boldsymbol{f}}^*$ on $\widehat{L}_{\text{P}}$ using Proposition 2. That is, if we directly optimize $L_{\text{P}}$ from a randomly initial point, we could cost more time on gradient descent for reaching $\boldsymbol{f}^*$. Since $\widehat{\boldsymbol{f}}^*$ is close to $\boldsymbol{f}^*$, it should be more efficient to use $\widehat{\boldsymbol{f}}^*$ as the initial guess for optimizing $\boldsymbol{f}^*$ in the quasi-Newton optimization procedure on $L_{\text{P}}$ (Jin & Mokhtari, 2021; 2023; Rodomanov & Nesterov, 2021) and we regard it as ETM-Refine method. In summary, one can utilize ETM-based approach (e.g., ETM-Exact, ETM-Approx and ETM-Refine) to transform SemiUOT into classic optimal transport problem. We also summarize the optimization details in Appendix C.

**Equivalent Transformation Mechanism for UOT.** We have obtained the marginal probability of SemiUOT via tackling Proposition 1 with proposed ETM-based method. In this section, we will further extend our methods for solving the marginal probability on UOT which is also a commonly exist optimization problem. That is, we will generalize ETM-based method on solving UOT problem accordingly.

**Proposition 3.** (Principles of Equivalent Transformation Mechanism for UOT) *Given UOT with KL-Divergence $J_{\text{UOT}}$, its Fenchel-Lagrange multipliers form is given:*

$$\min_{\boldsymbol{u}, \boldsymbol{v}, \zeta} \left[ \tau_a \sum_{i=1}^M a_i e^{-\frac{u_i + \zeta}{\tau_a}} + \tau_b \sum_{j=1}^N b_j e^{-\frac{v_j - \zeta}{\tau_b}} \right] \quad (8)$$

$$s.t. \ u_i + v_j + s_{ij} = C_{ij}, \quad s_{ij} \geq 0.$$

*where $\boldsymbol{u}$, $\boldsymbol{v}$, $\boldsymbol{s}$ and $\zeta$ denotes Lagrange multipliers. Moreover, UOT problem can also be transformed into classic optimal transport as follows:*

$$\min_{\boldsymbol{\pi} \geq 0} \mathcal{J}_{\text{U}} = \langle \boldsymbol{C}, \boldsymbol{\pi} \rangle \ s.t. \begin{cases} \boldsymbol{\pi} \mathbf{1}_N = \boldsymbol{a} \odot \exp\left( -\frac{\boldsymbol{u}^* + \zeta^*}{\tau_a} \right) = \boldsymbol{\alpha} \\ \boldsymbol{\pi}^\top \mathbf{1}_M = \boldsymbol{b} \odot \exp\left( -\frac{\boldsymbol{v}^* - \zeta^*}{\tau_b} \right) = \boldsymbol{\beta} \end{cases}$$

$$(9)$$

*Note that when $\tau_a, \tau_b \to \infty$, the source and target marginal probability can be determined as $\boldsymbol{\pi} \mathbf{1}_N = \sqrt{\omega} \boldsymbol{a}$ and $\boldsymbol{\pi}^\top \mathbf{1}_M = \boldsymbol{b} / \sqrt{\omega}$ where $\omega = \langle \boldsymbol{b}, \mathbf{1}_N \rangle / \langle \boldsymbol{a}, \mathbf{1}_M \rangle$ respectively.*

The proof of Proposition 3 can be found in Appendix D. Likewise, we set $v_j = \inf_{k \in [M]} (C_{kj} - u_k)$ according to the $c$-transform theorem (Villani et al., 2009) to simplify the calculation. Therefore we can obtain *Exact UOT Equation*:

$$\min_{\boldsymbol{u}, \zeta} L_{\text{U}} = \tau_a \sum_{i=1}^M a_i e^{-\frac{u_i + \zeta}{\tau_a}} + \tau_b e^{\frac{\zeta}{\tau_b}} \sum_{j=1}^N b_j e^{\frac{\sup_{k \in [M]} (u_k - C_{kj})}{\tau_b}}.$$

$$(10)$$

We first fix $\zeta$ then adopting L-BFGS method to optimize $L_{\text{U}}$. Then we optimize $\zeta = \kappa[\log(\sum_{i=1}^M a_i \exp(-u_i^\ell / \tau_a)) - \log(\sum_{j=1}^N b_j \exp(-v_j^\ell / \tau_b))]$ at the $\ell$-th iteration where $v_j^\ell = \inf_{k \in [M]} (C_{kj} - u_k^\ell)$ and $\kappa = \tau_a \tau_b / (\tau_a + \tau_b)$ by considering $\nabla_\zeta L_{\text{U}} = 0$. Here we regard the above process as the ETM-Exact approach for solving UOT problem. Note that the non-smooth function $\sup(\cdot)$ will result in inefficient optimization. However, if we directly apply the similar function approximation to replace $\sup(\cdot)$ following Eq.(4), the optimization problem becomes quite complex, making it relatively difficult to determine the iterative solutions. Meanwhile Proposition 2 enlightens us with a completely new ETM-Approx approach for optimizing UOT.

**Optimization 1.** (Calculation of ETM-Approx approach for UOT) Since the optimization problem in Eq.(8) is convex, we can also utilize block gradient descend to optimize the

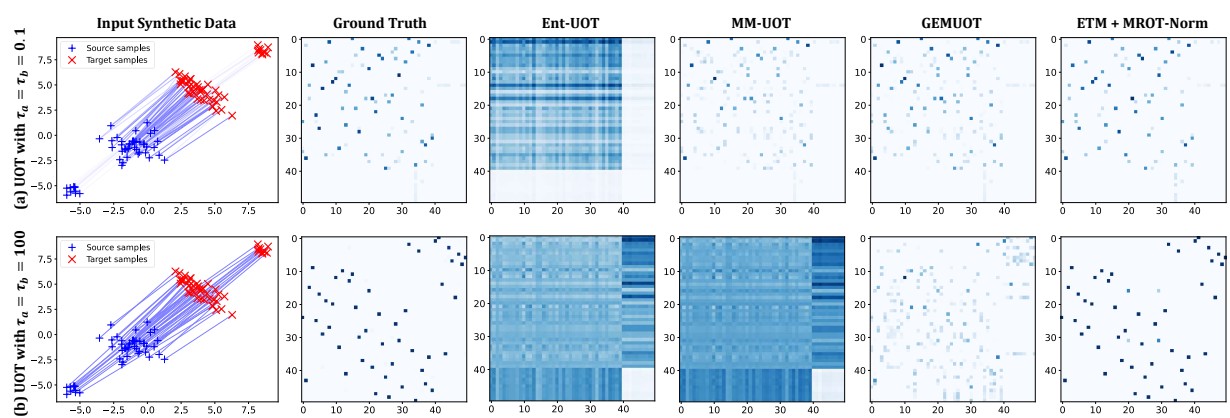

Figure 2. The UOT matching solutions on $\pi^*$ when $\tau_a = \tau_b = 0.1$ or $\tau_a = \tau_b = 100$ among the Ent-UOT (Pham et al., 2020), MM-UOT (Chapel et al., 2021), GEMUOT (Nguyen et al., 2023) and our proposed ETM + MROT-Norm with $\eta_G = 10^2$ and $\eta_{\text{Reg}} = 0.1$.

problem. Specifically, we first fix $\widehat{v}^l$ and optimize variable $\widehat{u}^l$ at the $l$-th iteration by replacing the original marginal probability $\boldsymbol{b}$ in Eq.(4) with $\boldsymbol{b} \odot \exp(-(\widehat{\boldsymbol{v}} - \zeta)/\tau_b) = \boldsymbol{\beta}$ accordingly to transform UOT into SemiUOT problem:

$$\min_{\widehat{\boldsymbol{u}}} \widehat{L}_{\text{U}}^u = \tau_a \sum_{i=1}^{M} a_i e^{-\frac{\widehat{u}_i + \zeta}{\tau_a}} + \sum_{j=1}^{N} \beta_j \left[ \log \left[ \sum_{k=1}^{M} e^{\frac{\widehat{u}_k - C_{kj}}{\epsilon}} \right]^\epsilon \right] + \zeta . \tag{11}$$

Note that it is equivalent to solve the following equation by taking the differentiation w.r.t. on $\widehat{u}_s$ over $\widehat{L}_{\text{U}}^u$ and set it 0:

$$\frac{\partial \widehat{L}_{\text{U}}^u}{\partial \widehat{u}_s} = -a_s e^{-\frac{\widehat{u}_s + \zeta}{\tau_a}} + e^{\frac{\widehat{u}_s}{\epsilon}} \sum_{j=1}^{N} \left[ \frac{\beta_j \exp\left(-\frac{C_{sj}}{\epsilon}\right)}{\sum_{k=1}^{M} \exp\left(\frac{\widehat{u}_k - C_{kj}}{\epsilon}\right)} \right] = 0. \tag{12}$$

Obviously, it is equivalent to replace $\boldsymbol{b}$ with $\boldsymbol{\beta}$ in Eq.(5) for solving Eq.(12). Then we can utilize the iteration step shown in Eq.(6) to obtain $\widehat{\boldsymbol{u}}^{l+1}$. After that we fix $\widehat{\boldsymbol{u}}^{l+1}$ and optimize variable $\widehat{\boldsymbol{v}}^{l+1}$ via $\widehat{v}_j^{l+1} = -\epsilon \log[\sum_{k=1}^{M} \exp((\widehat{u}_k^{l+1} - C_{kj})/\epsilon)]$. We can achieve the optimal solution on $\widehat{\boldsymbol{u}}^*$ and $\widehat{\boldsymbol{v}}^*$ via iteratively computing via the above procedure accordingly. Finally, we update variable $\zeta$ via considering $\zeta = (\tau_a \tau_b/(\tau_a + \tau_b))[\log(\sum_{i=1}^{M} a_i \exp(-\widehat{u}_i^*/\tau_a)) - \log(\sum_{j=1}^{N} b_j \exp(-\widehat{v}_j^*/\tau_b))]$. Due to the space limits, the deduction details are provided in Appendix E.

In summary, Optimization 1 for solving the UOT can be seen as an extension of Proposition 2 applied to SemiUOT, demonstrating the robust generalization capability of the proposed ETM method. Likewise, one can utilize $\widehat{\boldsymbol{u}}^*$ and $\widehat{\boldsymbol{v}}^*$ as the initial guess for solving Exact UOT Equation on Eq.(10) as ETM-Refine. Hence, UOT can be transformed into classic optimal transport using the ETM-based method.

### 3.2. KKT-Multiplier Regularization

According to the Proposition 1-3 that discussed in Section 3.1, we have figured out the marginal probability distributions on both UOT and SemiUOT with commonly used KL Divergence via proposed ETM-based method. Motivated by this, we can observe that the core mechanism of

UOT/SemiUOT is carefully reweighting the weights of different samples accordingly. If the samples are noise or outliers, the corresponding weights will be much smaller than the corresponding weights among similar data samples. Therefore, UOT/SemiUOT has better adaptability than traditional OT that commonly treats all data samples equally. In this section, we will further exploit the matching results of $\pi$ for SemiUOT and UOT using the following corollary:

**Corollary 1.** *Given any UOT/SemiUOT with KL divergence, we can transfer the original problem into classical optimal transport via adopting proposed ETM approach flexibly. We can further utilize existing OT solver for solving $\pi^*$ as:*

$$(\text{UOT}, \text{SemiUOT}) \xrightarrow{\text{ETM Method}} \text{OT} \xrightarrow{\text{OT Solver}} \pi^*.$$

This observation provides us with entirely new unified insights into solving the matching results of $\pi^*$ for UOT and SemiUOT. It is essential to utilize the proposed ETM-based method, as it offers a variety of OT solvers that yield more efficient and accurate results than directly optimizing UOT or SemiUOT. Specifically, one can further adopt Sinkhorn (Cuturi, 2013; Carlier, 2022), $\ell_2$-norm term (Blondel et al., 2018) or some other sparsification OT solver (Liu et al., 2023; Genevay et al., 2016) with different regularization terms to achieve the transportation solution on $\pi^*$.

Although previous efficient OT solvers (e.g., Sinkhorn (Cuturi, 2013)) could figure out $\pi^*$ efficiently, they often produce ambiguous matching results that may deviate significantly from the correct solutions (Montesuma et al., 2023; Liu et al., 2023). Therefore, finding an accurate matching solution for $\pi^*$ efficiently remains a challenge. Recalling the whole process of ETM-based approach, we not only obtain the marginal probabilities for each data sample, but also derive the multipliers $\boldsymbol{s}$ which can be further utilized.

**Corollary 2.** *Given the optimal $\boldsymbol{u}^*$ and $\boldsymbol{v}^*$ in UOT via ETM-based approach, one can obtain $\boldsymbol{s}$ on UOT by $s_{ij} = \max(0, C_{ij} - u_i^* - v_j^*)$. Likewise, the multipliers $\boldsymbol{s}$ on SemiUOT can be obtained via ETM-based approach as $s_{ij} = \max(0, C_{ij} - f_i^* - g_j^*)$. Multipliers $\boldsymbol{s}$ indicates the*

*value of $\boldsymbol{\pi}$, i.e., (case 1) $s_{ij} > 0$ when $\pi_{ij} = 0$ and (case 2) $s_{ij} = 0$ when $\pi_{ij} > 0$ according to the KKT conditions.*

The Corollary 2 demonstrates that the value of $\pi_{ij}$ can be reflected via $s_{ij}$. This observation inspires us to further utilize such useful information in calculating $\boldsymbol{\pi}^*$.

**Proposition 4.** (The Definition and Usage of KKT-Multiplier Regularization) *Given any OT with multiplier $\boldsymbol{s}$, one can obtain accurate solution $\boldsymbol{\pi}^*$ via proposed KKT-multiplier regularization term $\mathcal{G}(\boldsymbol{\pi}, \boldsymbol{s}) = \langle \boldsymbol{\pi}, \boldsymbol{s} \rangle$, which formulates Multiplier Regularized Optimal Transport (MROT):*

$$\min_{\boldsymbol{\pi} \geq 0} \mathcal{J}_{\mathrm{G}} = \langle \boldsymbol{C}, \boldsymbol{\pi} \rangle + \eta_G \langle \boldsymbol{\pi}, \boldsymbol{s} \rangle + \eta_{\mathrm{Reg}} \mathcal{L}_{\mathrm{Reg}}(\boldsymbol{\pi})$$
$$s.t. \ \boldsymbol{\pi} \mathbf{1}_N = \boldsymbol{\alpha}, \quad \boldsymbol{\pi}^\top \mathbf{1}_M = \boldsymbol{\beta}, \tag{13}$$

*where $\mathcal{L}_{\mathrm{Reg}}(\boldsymbol{\pi})$ denotes the regularization term on $\boldsymbol{\pi}$. $\boldsymbol{\alpha}$, $\boldsymbol{\beta}$ denote the final marginal probabilities obtained by ETM-based approach and $\eta_{\mathrm{Reg}}$, $\eta_G$ denotes the hyper parameter. Ideally, $\eta_G$ should be set as a relatively large number. Meanwhile the dual form of MROT is given as:*

$$\max_{\boldsymbol{\psi}, \boldsymbol{\phi}} L_{\mathrm{G}} = \langle \boldsymbol{\alpha}, \boldsymbol{\psi} \rangle + \langle \boldsymbol{\beta}, \boldsymbol{\phi} \rangle - \eta_{\mathrm{Reg}} \mathcal{L}_{\mathrm{Reg}}^* \left( \frac{\psi_i + \phi_j - \widetilde{C}_{ij}}{\eta_{\mathrm{Reg}}} \right) \tag{14}$$

*where $\widetilde{C}_{ij} = C_{ij} + \eta_G s_{ij}$ and $\boldsymbol{\phi}$ and $\boldsymbol{\psi}$ denote the Lagrange multipliers for MROT. $\mathcal{L}_{\mathrm{Reg}}^*(\cdot)$ denotes the conjugate function of $\mathcal{L}_{\mathrm{Reg}}(\cdot)$ and one can figure out the matching results of $\boldsymbol{\pi}$ via solving $\nabla_{\pi_{ij}} \mathcal{L}_{\mathrm{Reg}}(\pi_{ij}) = (\psi_i + \phi_j - \widetilde{C}_{ij})/\eta_{\mathrm{Reg}}$.*

The deduction process of MROT can be found in Appendix G. That is, minimizing $s_{ij}\pi_{ij}$ to 0 could result in $s_{ij}\pi_{ij} = 0$ which is consistent with the KKT complementary condition. Generally, we can adopt different kinds of regularization term $\mathcal{L}_{\mathrm{Reg}}(\cdot)$ on MROT for optimization. For instance, one can use the widely adopted entropy regularization term $\mathcal{L}_{\mathrm{Reg}}(\boldsymbol{\pi}) = -\langle \boldsymbol{\pi}, \log(\boldsymbol{\pi}) - 1 \rangle$ to formulate Entropic Multiplier Regularized Optimal Transport (MROT-Ent). The matching results of $\boldsymbol{\pi}$ in MROT-Ent can be obtained as:

$$\pi_{ij} = \exp\left( -\frac{\eta_G s_{ij}}{\eta_{\mathrm{Reg}}} \right) \exp\left( \frac{\psi_i + \phi_j - C_{ij}}{\eta_{\mathrm{Reg}}} \right) \tag{15}$$

Obviously, the multipliers information $\boldsymbol{s}$ has been involved for achieving more accurate solutions. Specifically, the non-matching samples pairs will get lower value on $\pi_{ij}$ since $\mathcal{G}(\boldsymbol{\pi}, \boldsymbol{s}) = \langle \boldsymbol{\pi}, \boldsymbol{s} \rangle$ avoids rigorous results. Otherwise, the matching results on $\pi_{ij}$ will mainly be determined by the transportation cost. Similarly, one can also adopt $L_2$-norm regularization term $\mathcal{L}_{\mathrm{Reg}}(\boldsymbol{\pi}) = \frac{1}{2} \langle \boldsymbol{\pi}, \boldsymbol{\pi} \rangle$ to formulate Sparse Multiplier Regularized Optimal Transport (MROT-Norm) with similar characteristics. More discussions on MROT-Norm can be found in Appendix. In conclusion, we can integrate the ETM-based approach with MROT method to solve UOT and SemiUOT, achieving accurate results for both marginal probabilities and the matching solution $\pi_{ij}$.

## 4. Related Works

**Unbalanced and Semi-Unbalanced Optimal Transport.** (1) *Related works on UOT*: UOT with KL divergence has been widely investigated for dealing with diverse applications (Peyré et al., 2019; De Plaen et al., 2023; Séjourné et al., 2019; Le et al., 2022). Different types of UOT solutions can be distinguished in terms of using entropy regularization term or not. Involving entropy in UOT can enhance the model scalability, yet resulting in dense matching results (Sinkhorn & Knopp, 1967; Balaji et al., 2020). Latest, (Chapel et al., 2021) further considers UOT without entropy terms by Majorization-Minimization (MM) (Chizat et al., 2018; Sun et al., 2016) or regularization path methods (Mairal & Yu, 2012; Massias et al., 2018; Liu & Nocedal, 1989). However, the nature of MM algorithm inherits inexact proximal point of KL term (Xie et al., 2020), which still causes dense mapping when $\tau$ becomes larger. Meanwhile regularization path methods could be quite slow in computation especially when $\tau \to +\infty$. Furthermore, as the number of samples increases, it can lead to high storage space consumption which can be problematic. (2) *Related works on SemiUOT*: SemiUOT with KL divergence only relaxes one of the marginal constraints comparing with UOT. (Le et al., 2021) first fully investigated the corresponding problem and proposed Robust-SemiSinkhorn algorithm. Nevertheless, it still suffers from inaccurate matching solutions with entropy regularization term. Currently, there only exists extremely few works for solving SemiUOT (Montesuma et al., 2024). Therefore, how to efficiently provide accurate solution on both UOT and SemiUOT is still a challenging problem.

## 5. Experiments

In this section, we conduct experiments on both synthetic and real-world datasets to evaluate our proposed methods.

### 5.1. Experimental setup

**Synthetic Datasets.** We first conduct the experiments on the synthetic datasets. That is, we set the source and target domain distributions as $\mathbb{P}_X = \mathcal{N}\left( \begin{bmatrix} -1 \\ -1 \end{bmatrix}, \begin{bmatrix} 1 & 0 \\ 0 & 1 \end{bmatrix} \right)$ and $\mathbb{P}_Z = \mathcal{N}\left( \begin{bmatrix} 4 \\ 4 \end{bmatrix}, \begin{bmatrix} 1 & -0.8 \\ -0.8 & 1 \end{bmatrix} \right)$ following previous works (Flamary et al., 2021; Chapel et al., 2021). We will sample a number of source and target data via $\boldsymbol{x} \sim \mathbb{P}_X$ and $\boldsymbol{z} \sim \mathbb{P}_Z$ respectively to establish the synthetic datasets.

**Datasets for Domain Adaptation.** We conduct the unsupervised domain adaptation tasks on *Digits* (Lecun et al., 1998; Hull, 2002; Netzer et al., 2011), *Office-Home* (Venkateswara et al., 2017), and *VisDA* (Peng et al., 2018). More details on these datasets are provided in Appendix H.

**Baselines.** We compare ETM-Refine with MGOT method with the following state-of-the-art UOT/SemiUOT solvers on the synthetic datasets. (1) **Ent-UOT** (Pham et al., 2020)

*Table 1.* Classification accuracy (%) on *Office-Home* for UDA and Partial UDA

| Method for Partial UDA | Ar→Cl | Ar→Pr | Ar→Rw | Cl→Ar | Cl→Pr | Cl→Rw | Pr→Ar | Pr→Cl | Pr→Rw | Rw→Ar | Rw→Cl | Rw→Pr | Avg |
|---|---|---|---|---|---|---|---|---|---|---|---|---|---|
| ResNet (He et al., 2016) | 46.3 | 67.5 | 75.9 | 59.1 | 59.9 | 62.7 | 58.2 | 41.8 | 74.9 | 67.4 | 48.2 | 74.2 | 61.4 |
| ETN (Cao et al., 2019) | 59.2 | 77.0 | 79.5 | 62.9 | 65.7 | 75.0 | 68.3 | 55.4 | 84.4 | 75.7 | 57.7 | 84.5 | 70.5 |
| JUMBOT (Fatras et al., 2021) | 62.7 | 77.5 | 84.4 | 76.0 | 73.3 | 80.5 | 74.7 | 60.8 | 85.1 | 80.2 | 66.5 | 83.9 | 75.5 |
| AR (Gu et al., 2021) | 67.4 | 85.3 | 90.0 | 77.3 | 70.6 | 85.2 | 79.0 | 64.8 | 89.5 | 80.4 | 66.2 | 86.4 | 78.3 |
| m-POT (Nguyen et al., 2022) | 64.6 | 80.6 | 87.2 | 76.4 | 77.6 | 83.6 | 77.1 | 63.7 | 87.6 | 81.4 | 68.5 | 87.4 | 78.0 |
| MOT (Luo & Ren, 2023) | 63.1 | 86.1 | 92.3 | 78.7 | 85.4 | 89.6 | 79.8 | 62.3 | 89.7 | 83.8 | 67.0 | 89.6 | 80.6 |
| MOT + UOT(ETM + MROT-Ent) | 65.2 | 87.3 | 92.8 | 79.5 | 86.4 | 91.0 | 80.8 | 64.5 | 90.7 | 84.5 | 67.9 | 90.4 | 81.8 |
| MOT + UOT(ETM + MROT-Norm) | 65.8 | 88.0 | 93.1 | 79.9 | 86.2 | 91.3 | 81.4 | 64.9 | 91.2 | 84.9 | 68.3 | 90.7 | 82.1 |
| MOT + SemiUOT(Robust-SemiSinkhorn) | 66.0 | 88.2 | 93.0 | 80.5 | 86.8 | 91.5 | 81.3 | 65.2 | 91.6 | 85.2 | 68.5 | 90.9 | 82.4 |
| MOT + SemiUOT(ETM + MROT-Ent) | 68.6 | 90.4 | 94.2 | 83.7 | 89.5 | 93.9 | 83.5 | 65.4 | 93.9 | 88.4 | **71.8** | 92.1 | 84.8 |
| MOT + SemiUOT(ETM + MROT-Norm) | **69.1** | **90.7** | **94.6** | **84.0** | **90.3** | **94.0** | **83.8** | **67.9** | **94.4** | **88.5** | 71.3 | **93.6** | **85.2** |
| **Method for UDA** | Ar→Cl | Ar→Pr | Ar→Rw | Cl→Ar | Cl→Pr | Cl→Rw | Pr→Ar | Pr→Cl | Pr→Rw | Rw→Ar | Rw→Cl | Rw→Pr | Avg |
| ResNet (He et al., 2016) | 34.9 | 50.0 | 58.0 | 37.4 | 41.9 | 46.2 | 38.5 | 31.2 | 60.4 | 53.9 | 41.2 | 59.9 | 46.1 |
| DeepJDOT (Damodaran et al., 2018) | 50.7 | 68.6 | 74.4 | 59.9 | 65.8 | 68.1 | 55.2 | 46.3 | 73.8 | 66.0 | 54.9 | 78.3 | 63.5 |
| ROT (Balaji et al., 2020) | 47.2 | 71.8 | 76.4 | 58.6 | 68.1 | 70.2 | 56.5 | 45.0 | 75.8 | 69.4 | 52.1 | 80.6 | 64.3 |
| JUMBOT (Fatras et al., 2021) | 55.2 | 75.5 | 80.8 | 65.5 | 74.4 | 74.9 | 65.2 | 52.7 | 79.2 | 73.0 | 59.9 | 83.4 | 70.0 |
| JUMBOT + UOT(MMUOT) | 56.3 | 76.2 | 81.6 | 66.0 | 75.3 | 75.1 | 66.4 | 52.9 | 79.2 | 73.8 | 60.7 | 84.1 | 70.6 |
| JUMBOT + UOT(GEMUOT) | 57.5 | 77.4 | 82.7 | 67.2 | 76.0 | 75.6 | 66.1 | 54.5 | 80.5 | 74.9 | 61.8 | 85.2 | 71.6 |
| JUMBOT + UOT($\ell_2$-Norm Solver) | 57.0 | 76.7 | 81.8 | 66.1 | 75.5 | 75.5 | 65.9 | 53.4 | 79.6 | 74.2 | 60.6 | 83.3 | 70.7 |
| JUMBOT + UOT(Sparse Solver) | 57.8 | 77.1 | 82.3 | 66.7 | 76.2 | 75.8 | 67.0 | 54.1 | 80.7 | 75.4 | 61.3 | 84.6 | 71.5 |
| JUMBOT + UOT(ETM + MROT-Ent) | 59.0 | 78.5 | 83.4 | **68.7** | 77.1 | 77.6 | 68.3 | 57.2 | 82.4 | 76.2 | **62.5** | 86.4 | 73.1 |
| JUMBOT + UOT(ETM + MROT-Norm) | **59.4** | **78.7** | **84.1** | 68.5 | **77.3** | **78.5** | **68.6** | **57.9** | **82.8** | **76.3** | **62.5** | **86.5** | **73.4** |

*Table 2.* Classification accuracy (%) on Digits (Source: LeNet) and VisDA dataset (Source:ResNet50) for UDA task

| Method | S→M | M→U | U→M | Avg | VisDA |
|---|---|---|---|---|---|
| Source | 68.3±0.3 | 65.3±0.5 | 66.2±0.2 | 66.6 | 52.4 |
| DeepJDOT (Damodaran et al., 2018) | 95.4±0.1 | 95.6±0.4 | 96.4±0.3 | 95.8 | 68.0 |
| JUMBOT (Fatras et al., 2021) | 98.9±0.1 | 96.7±0.5 | 98.2±0.1 | 97.9 | 72.5 |
| JUMBOT + UOT(ETM + MROT-Ent) | 99.4±0.1 | 98.7±0.3 | 99.2±0.1 | 99.1 | 73.6 |
| JUMBOT + UOT(ETM + MROT-Norm) | 99.7±0.1 | 99.3±0.2 | 99.6±0.1 | **99.5** | **74.2** |

utilizes the entropy regularization term on tackling UOT problem. (2) **MMUOT** (Chapel et al., 2021) adopts majority maximization algorithm for solving UOT. (3) **GEMUOT** (Nguyen et al., 2023) adopts $\ell_2$-norm term for reaching transport solutions on UOT which is the state-of-the-art approach. (4) **Robust-SemiSinkhorn** (Le et al., 2021) adopts the entropy regularization term for solving SemiUOT problem. We also involve **DeepJDOT** (Damodaran et al., 2018), **ROT** (Balaji et al., 2020), **JUMBOT** (Fatras et al., 2021), **ETN** (Cao et al., 2019), **AR** (Gu et al., 2021), **m-POT** (Nguyen et al., 2022), **MOT** (Luo & Ren, 2023) as the model baselines for the domain adaptation task. The model details will be provided in Appendix.I.

**Implemented details.** For both synthetic and real-world datasets, we set $\epsilon = 0.01$ on both $\widehat{L}_U$ and $\widehat{L}_P$. We set $\eta_G = 10^2$ and $\eta_{\text{Reg}} = 0.1$ for MROT in the calculation. The initial value of $\widehat{u}^{(0)}$ and $\widehat{f}^{(0)}$ as set as zero vectors. The initial sample weights are set to be equal, i.e., $a_i = \frac{1}{M}$ and $b_j = \frac{1}{N}$. And we adopt square Euclidean distance for the cost $C_{ij}$. Besides, we follow the same framework and experimental settings as UDA model **JUMBOT** (Fatras et al., 2021) for domain adaptation. Meanwhile, we adopt the same framework and experimental settings as partial UDA model **MOT** (Luo & Ren, 2023) for partial domain adaptation. For all the experiments, we perform five random experiments and report the average results.

### 5.2. Performance on Synthetic and Real-World Datasets

**Performance on Synthetic Datasets.** We sample 50 data samples on both source and target distributions for finding $\boldsymbol{\pi}^*$ on UOT/SemiUOT. We first set $\tau = \{0.1, 100\}$ on SemiUOT and the matching solutions are shown in Fig.1(a)-(b). Note that we randomly sample 20% of noise in the source datasets. We can observe that previous method **Robust-SemiSinkhorn** could lead to ambiguious results. Our proposed ETM-Refine with MROT+Ent can reach relatively clear results even if $\tau$ is large (e.g., $\tau = 100$). More importantly, ETM-Refine with MROT+Norm can achieve more precise results comparing with ETM-Refine with MROT+Ent shown in Fig.1. Then we also set $\tau_a = \tau_b = \{0.1, 100\}$ on UOT and the matching solutions are shown in Fig.2(a)-(b). From that we can observe: (1) **Ent-UOT** could merely provide dense transport solutions which are inaccurate. (2) **MMUOT** obtains relatively accurate solutions when $\tau$ is small. However, **MMUOT** cannot better handle the case when $\tau$ is large (e.g., $\tau = 100$) due to the deterioration of majority maximization algorithm. (3) **GEMUOT** can even reach more sparse matching solution against **Ent-UOT** and **MMUOT** with the aid of $\ell_2$-norm term. However, the matching results obtained from **GEMUOT** remain coarse and ambiguous, especially when $\tau$ is large. (4) ETM-Refine with MGOT-Norm can reach more accurate results with a smaller error compared to the standard UOT solutions.

**Performance on Real-World Datasets.** We further conduct the experiments on the real-world datasets to validate the our proposed method. The experimental UDA task results on *Office-Home*, *Digits* and *VisDA* are shown in Table.1 and Table.2. We also directly adopt $\ell_2$-norm (Blondel et al., 2018) and sparse solver (Liu et al., 2023) on solving $\mathcal{J}_U$. We can observe that replacing entropy-based UOT with other regularization term (e.g., **GEMUOT** or $\mathcal{J}_U$ with $\ell_2$-norm) could lead to better results. Moreover, our proposed ETM-Refine with MROT obtains the best performance, which indicates the method efficacy for finding more accurate results. Then we adopt the same experimental protocol as **MOT** to establish the partial UDA task where target label space is a subset of source label space and it is more challenging than classic UDA task (Cao et al., 2018; Luo et al., 2020). The

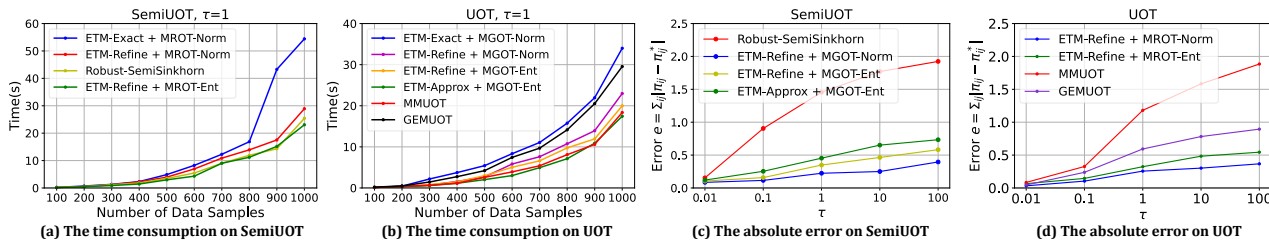

*Figure 3.* The time consumption and computation error analysis $e = \sum_{ij} ||\pi_{ij} - \pi_{ij}^*||_1$ on UOT and SemiUOT. (Best viewed in color)

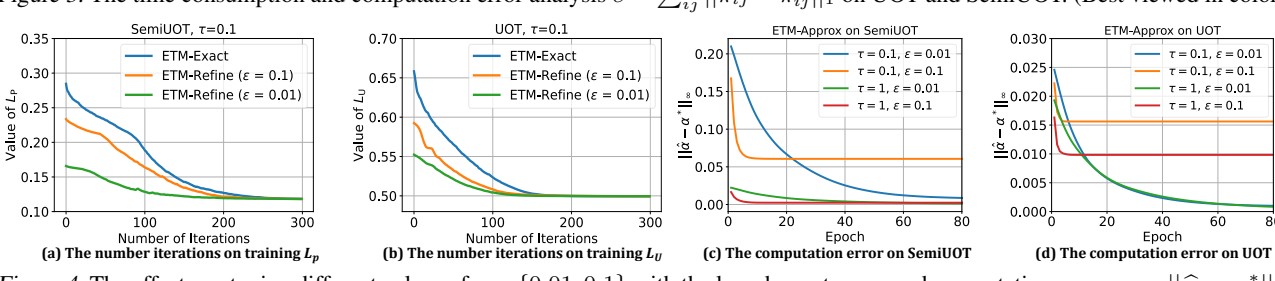

*Figure 4.* The effects on tuning different values of $\epsilon = \{0.01, 0.1\}$ with the loss descent curve and computation error $e_\alpha = ||\widehat{\alpha} - \alpha^*||_\infty$ for the proposed ETM-Approx methods on solving SemiUOT and UOT problems.

partial UDA results on *Office-Home* are also shown in Table.1. We can easily observe that MOT + SemiUOT (ETM + MROT-Norm) achieves the best performance on the partial UDA task. UOT relaxes the dual transportation constraints, thus resulting in that some target samples cannot be transported to the source domain. Meanwhile, SemiUOT could overcome the mentioned issue while avoid negative transfer in partial UDA, which boosts the model performance. We also conduct more experiments with applications and please kindly refer to Appendix.I, J, K for more details.

### 5.3. Analysis

**Solver Comparison.** To further analyses the proposed ETM-based method with MROT, we conduct the solver comparison on the aspects of *computation time*, *computation error*, and *model performance on real-world datasets*. We first sample the same number of source/target data samples from $\mathbb{P}_X$ and $\mathbb{P}_Z$ respectively. Then we conduct the experiments on both UOT and SemiUOT with $\tau_a = \tau_b = \tau = 1$ and the results are shown in Fig.3(a)-(b). We can conclude that ETM-Exact with MROT-Norm is most time-consuming. Meanwhile ETM-Refine reaches similar computation time with ETM-Approx suggests that utilizing $\widehat{u}^*$ or $\widehat{f}^*$ could accelerate the process for finding $u^*$ or $f^*$ via L-BFGS algorithm. Moreover, we calculate the computation error $e$ between matching solution $\pi$ learned by ETM with MROT and the standard UOT/SemiUOT solution with CVXPY as $\pi^*$ via absolute error $e = \sum_{i,j} ||\pi_{ij} - \pi_{ij}^*||_1$. We sample 500 number of data samples ranging from $\tau_a = \tau_b = \tau = \{0.01, 0.1, 1, 10, 100\}$ for calculation and the results are shown in Fig.3(c)-(d). We can observe that although ETM-Approx with MROT-Ent has the fastest computation speed, the provided results $\pi$ still have the highest error compared to the ground truth $\pi^*$. Meanwhile ETM-Refine with MROT-Norm can further reach more accurate

solutions against MROT-Ent. We also collect the computation error on UOT using **MMUOT** and **GEMUOT**. We can observe that our propose ETM-Refine method achieves much better results, especially when $\tau$ is relatively large, which is consistent with our discovery in Fig.1-Fig.2.

**Parameter sensitivity.** We finally study the effects of hyper-parameters on model performance. We tune $\epsilon$ in range of $\epsilon \in \{0.01, 0.1\}$ and show the results in Fig.4(a)-(d). We can observe that smaller $\epsilon$ could provide good approximation on UOT/SemiUOT, reducing the iteration steps for optimizing $L_U$ and $L_P$. Although $\epsilon$ could hardly effect the performance on ETM-Refine, larger value on $\epsilon$ could consume more iteration steps for solving $L_U$ and $L_P$ since the initial values are less accurate. Additionally, we collect the computation error $e_\alpha = ||\widehat{\alpha} - \alpha^*||_\infty$, which measures the discrepancy between the marginal probability learned via ETM-Approx $\widehat{\alpha}$ and the ground truth $\alpha^*$. Larger values of $\epsilon$ may fail to reduce the computation error $e_\alpha$ when compared to smaller values of $\epsilon$. Hence we set $\epsilon = 0.01$ empirically and more experimental results can be found in Appendix.L, M.

### 6. Conclusion

In this paper, we propose Equivalent Transformation Mechanism (ETM) approach with ETM-Exact, ETM-Approx and ETM-Refine to solve the marginal probabilities of SemiUOT and UOT. We illustrate that the essence of UOT/SemiUOT is reweighting data samples accordingly and thus UOT/SemiUOT problem can be transformed into standard optimal transport. Moreover, we propose KKT-Multiplier Regularization with Multiplier Regularized Optimal Transport (MROT) to obtain more accurate solutions. We conduct experiments to demonstrate the superior performance of our proposed ETM with MROT, on both synthetic and real-world datasets of different tasks and applications.

## 7. Potential Broader Impact

This paper provides a new insight on solving (semi) unbalanced optimal transport problem. Moreover, we first utilize multipliers information into solving transportation solutions. The extensive experiments on both real-world and synthetic datasets with diverse domain adaptation problems show the efficacy of the proposed ETM method with MROT.

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

# Appendix

## A. Proof of Proposition 1

**Proposition 1.** (Principles of Equivalent Transformation Mechanism for SemiUOT) *Given SemiUOT with KL-Divergence* $J_{\text{SemiUOT}}$, *one can obtain its Fenchel-Lagrange multipliers form as:*

$$\min_{\boldsymbol{f}, \boldsymbol{g}, \zeta} \left[ \tau \sum_{i=1}^{M} a_i e^{-\frac{f_i + \zeta}{\tau}} - \sum_{j=1}^{N} b_j (g_j - \zeta) \right] \tag{16}$$

$$s.t. \ f_i + g_j + s_{ij} = C_{ij}, \quad s_{ij} \geq 0.$$

*where* $\boldsymbol{f}$, $\boldsymbol{g}$, $\boldsymbol{s}$ *and* $\zeta$ *denotes Lagrange multipliers. Moreover, SemiUOT problem can be transformed into classic optimal transport as follows:*

$$\min_{\boldsymbol{\pi} \geq 0} \mathcal{J}_{\text{P}} = \langle \boldsymbol{C}, \boldsymbol{\pi} \rangle$$

$$s.t. \begin{cases} \boldsymbol{\pi} \mathbf{1}_N = \boldsymbol{a} \odot \exp \left( -\frac{\boldsymbol{f}^* + \zeta^*}{\tau} \right) = \boldsymbol{\alpha} \\ \boldsymbol{\pi}^\top \mathbf{1}_M = \boldsymbol{b} \end{cases} \tag{17}$$

*Note that when* $\tau \to \infty$, *the source marginal probability can be determined as* $\boldsymbol{\pi} \mathbf{1}_N = \omega \boldsymbol{a}$ *where* $\omega = \langle \boldsymbol{b}, \mathbf{1}_N \rangle / \langle \boldsymbol{a}, \mathbf{1}_M \rangle$.

*Proof.* To start with, we first review the definition of SemiUOT as shown below:

$$\min_{\pi_{ij} \geq 0} J_{\text{SemiUOT}} = \langle \boldsymbol{C}, \boldsymbol{\pi} \rangle + \tau \text{KL} (\boldsymbol{\pi} \mathbf{1}_N \| \boldsymbol{a})$$

$$s.t. \ \boldsymbol{\pi}^\top \mathbf{1}_M = \boldsymbol{b}. \tag{18}$$

Then we can rewrite the optimization problem:

$$\min_{\boldsymbol{\pi} \geq 0} J = \langle \boldsymbol{C}, \boldsymbol{\pi} \rangle + \tau \text{KL} (\boldsymbol{\pi} \mathbf{1}_N \| \boldsymbol{a})$$

$$s.t. \begin{cases} (\text{Constraint}): \ \boldsymbol{\pi}^\top \mathbf{1}_M = \boldsymbol{b} \\ (\text{Optional}): \ \boldsymbol{\pi} \mathbf{1}_N = \boldsymbol{\alpha} \end{cases} \tag{19}$$

Note that we do not need to know the exact value of $\boldsymbol{\alpha}$ beforehand. We adopt this optional constraint only for simplifying the following deduction. The Lagrange multipliers of SemiUOT with KL-Divergence is given as:

$$\max_{\boldsymbol{s} \geq 0, \boldsymbol{f}, \boldsymbol{g}, \zeta} \min_{\boldsymbol{\pi} \geq 0} \mathcal{J}_{\text{SemiUOT}} = \tau \text{KL} (\boldsymbol{\pi} \mathbf{1}_N \| \boldsymbol{a}) + \langle \boldsymbol{f} + \zeta, \boldsymbol{\pi} \mathbf{1}_N \rangle + \langle \boldsymbol{g} - \zeta, \boldsymbol{b} \rangle +$$

$$\langle \boldsymbol{C} - \boldsymbol{u} \otimes \mathbf{1}_N^\top - \mathbf{1}_M \otimes \boldsymbol{v}^\top - \boldsymbol{s}, \boldsymbol{\pi} \rangle \tag{20}$$

where $\boldsymbol{f}$, $\boldsymbol{g}$, $\boldsymbol{s}$ and $\zeta$ are dual variables. By taking the differentiation on $\pi_{ij}$ we have:

$$\frac{\partial \mathcal{J}_{\text{SemiUOT}}}{\partial \pi_{ij}} = \left[ \tau \log \frac{\sum_{j=1}^{N} \pi_{ij}}{a_i} + f_i + \zeta \right] + (C_{ij} - f_i - g_j - s_{ij})$$

$$= C_{ij} + \tau \log \frac{\sum_{j=1}^{N} \pi_{ij}}{a_i} + \zeta - g_j - s_{ij} \tag{21}$$

$$= 0$$

Therefore we can obtain the results as:

$$\begin{cases} \sum_{j=1}^{N} \pi_{ij} = a_i \exp \left( -\frac{f_i + \zeta}{\tau} \right) \\ \sum_{i=1}^{M} \pi_{ij} = b_j \\ C_{ij} - f_i - g_j - s_{ij} = 0 \end{cases} \tag{22}$$

After that, we can take these back into KL-Divergence to simplify the calculation:

$$\tau \mathrm{KL}\left(\boldsymbol{\pi}\mathbf{1}_N \| \boldsymbol{a}\right) + \langle \boldsymbol{f} + \zeta, \boldsymbol{\pi}\mathbf{1}_N \rangle$$

$$= \tau \mathrm{KL}\left(\left\langle \boldsymbol{a}, \exp\left(-\frac{\boldsymbol{f}+\zeta}{\tau}\right)\right\rangle \| \boldsymbol{a}\right) + \left\langle \boldsymbol{f}+\zeta, \boldsymbol{a}\exp\left(-\frac{\boldsymbol{f}+\zeta}{\tau}\right)\right\rangle$$

$$= \tau \sum_{i=1}^{M}\left[a_i \exp\left(-\frac{f_i+\zeta}{\tau}\right)\log\frac{a_i\exp\left(-\frac{f_i+\zeta}{\tau}\right)}{a_i} - a_i\exp\left(-\frac{f_i+\zeta}{\tau}\right) + a_i\right] + \sum_{i=1}^{M}(f_i+\zeta)a_i\exp\left(-\frac{f_i+\zeta}{\tau}\right) \quad (23)$$

$$= \sum_{i=1}^{M}\left[-\tau a_i \exp\left(-\frac{f_i+\zeta}{\tau}\right) + \tau a_i\right]$$

Therefore we can obtain its Fenchel-Lagrange multipliers form of SemiUOT as:

$$\min_{\boldsymbol{f},\boldsymbol{g},\zeta} \mathcal{J}_{\mathrm{SemiUOT}} = -\tau\mathrm{KL}\left(\boldsymbol{\pi}\mathbf{1}_N\|\boldsymbol{a}\right) - \langle\boldsymbol{f}+\zeta, \boldsymbol{\pi}\mathbf{1}_N\rangle - \langle\boldsymbol{g}-\zeta, \boldsymbol{\pi}^\top\mathbf{1}_M\rangle$$

$$= \tau\exp\left(-\frac{\zeta}{\tau}\right)\left\langle\boldsymbol{a}, \exp\left(-\frac{\boldsymbol{f}}{\tau}\right)\right\rangle - \langle\boldsymbol{g}-\zeta, \boldsymbol{b}\rangle + \mathcal{O}_{\mathrm{Const}} \quad (24)$$

$$s.t. \ f_i + g_j \le C_{ij}$$

where $\mathcal{O}_{\mathrm{Const}} = -\sum_{i=1}^{M}\tau a_i$ and we can neglect it during the following calculation. Once we obtain the optimal solution on $\boldsymbol{f}^*$, $\boldsymbol{g}^*$ and $\zeta^*$, we will discover that:

$$\tau\mathrm{KL}\left(\boldsymbol{\pi}\mathbf{1}_N\|\boldsymbol{a}\right) = \tau\mathrm{KL}\left(\left\langle\boldsymbol{a}, \exp\left(-\frac{\boldsymbol{f}^*+\zeta^*}{\tau}\right)\right\rangle \| \boldsymbol{a}\right) = \mathrm{Const} \quad (25)$$

Hence SemiUOT problem can be transformed into classic optimal transport accordingly. Finally we can obtain the optimal solution on $\zeta$ by considering $\frac{\partial\mathcal{J}_{\mathrm{SemiUOT}}}{\partial\zeta} = 0$ as below:

$$\zeta = \tau\left[\log\left(\sum_{i=1}^{M}a_i\exp\left(-\frac{f_i}{\tau}\right)\right) - \log\left(\sum_{j=1}^{N}b_j\right)\right]. \quad (26)$$

Once we set $\tau \to \infty$, the results of the limitation will be shown as:

$$\lim_{\tau\to+\infty}a_i\exp\left(-\frac{f_i+\zeta}{\tau}\right) = \lim_{\tau\to+\infty}a_i\exp\left(-\frac{\zeta}{\tau}\right) = a_i\frac{\langle\boldsymbol{b}, \mathbf{1}_N\rangle}{\langle\boldsymbol{a}, \mathbf{1}_M\rangle} = \omega a_i \quad (27)$$

Therefore we conclude the proof of the Proposition 1. $\qquad\square$

## B. Proof of Proposition 2

**Proposition 2.** (Calculation for Approximate SemiUOT Equation) *Given Approximate SemiUOT equation $\widehat{L}_{\mathrm{P}}$, it can be optimized via Equivalent Transformation Mechanism with Approximation (ETM-Approx). That is, ETM-Approx aims to solve the following equation for each $\widehat{f}_s$:*

$$\frac{\partial\widehat{L}_{\mathrm{P}}}{\partial\widehat{f}_s} = -a_s e^{-\frac{\widehat{f}_s+\zeta}{\tau}} + e^{\frac{\widehat{f}_s}{\epsilon}}\sum_{j=1}^{N}\left[\frac{b_j\exp\left(-\frac{C_{sj}}{\epsilon}\right)}{\sum_{k=1}^{M}\exp\left(\frac{\widehat{f}_k-C_{kj}}{\epsilon}\right)}\right] = 0. \quad (28)$$

*Specifically, we can adopt fixed-point iteration method for solving Eq.(28) at the $\ell$-th iteration as follows:*

$$\begin{cases} \widehat{f}_1^{\ell+1} = \nu\left[\log\left(a_1 e^{-\frac{\zeta}{\tau}}\right) - \log\left[\sum_{j=1}^{N}\left(\frac{b_j e^{-C_{1j}/\epsilon}}{\mathscr{W}_{\epsilon,j}(\widehat{\boldsymbol{f}}^\ell)}\right)\right]\right] \\ \vdots \\ \widehat{f}_M^{\ell+1} = \nu\left[\log\left(a_M e^{-\frac{\zeta}{\tau}}\right) - \log\left[\sum_{j=1}^{N}\left(\frac{b_j e^{-C_{Mj}/\epsilon}}{\mathscr{W}_{\epsilon,j}(\widehat{\boldsymbol{f}}^\ell)}\right)\right]\right], \end{cases} \quad (29)$$

*where $\nu = \tau\epsilon/(\tau+\epsilon)$ for simplification and $\mathscr{W}_{\epsilon,j}(\widehat{\boldsymbol{f}}^\ell)$ denotes the corresponding calculation as shown below:*

$$\mathscr{W}_{\epsilon,j}(\widehat{\boldsymbol{f}}^\ell) = \sum_{k=1}^{M}\exp\left(\frac{\widehat{f}_k^\ell - C_{kj}}{\epsilon}\right). \quad (30)$$

*The proposed procedure can be convergence with theoretical guarantee after $\mathcal{T}$-th inner iteration. Finally, updating the Lagrange multiplier $\zeta$ by considering $\nabla_\zeta \widehat{L}_P = 0$ via $\zeta = \tau[\log(\sum_{i=1}^M a_i \exp(-\widehat{f}_i/\tau)) - \log(\sum_{j=1}^N b_j)]$. One can achieve the optimal solution on $\widehat{\boldsymbol{f}}$ and $\zeta$ via iterative computing accordingly.*

*Proof.* We first review the proposed Approximate SemiUOT Equation $\widehat{L}_P$ as below:

$$\min_{\widehat{\boldsymbol{f}}, \zeta} \widehat{L}_P = \tau \sum_{i=1}^M a_i e^{-\frac{\widehat{f}_i + \zeta}{\tau}} + \sum_{j=1}^N b_j \left[ \epsilon \log \left[ \sum_{k=1}^M e^{\frac{\widehat{f}_k - C_{kj}}{\epsilon}} \right] + \zeta \right] \tag{31}$$

Then we consider optimizing $\widehat{f}_s$ as follows:

$$\frac{\partial \widehat{L}_P}{\partial \widehat{f}_s} = 0 \quad \Rightarrow \quad \exp \left( \frac{\tau + \epsilon}{\tau \epsilon} \widehat{f}_s \right) = \frac{a_s e^{-\frac{\zeta}{\tau}}}{\sum_{j=1}^N \left( \frac{b_j \exp(-C_{sj}/\epsilon)}{\mathscr{W}_{\epsilon,j}(\widehat{\boldsymbol{f}})} \right)} \tag{32}$$

At that time we adopt fixed-point iteration method to optimize $\widehat{\boldsymbol{f}}$ accordingly:

$$\begin{cases} \widehat{f}_1^{\ell+1} = \nu \left[ \log \left( a_1 e^{-\frac{\zeta}{\tau}} \right) - \log \left[ \sum_{j=1}^N \left( \frac{b_j e^{-C_{1j}/\epsilon}}{\mathscr{W}_{\epsilon,j}(\widehat{\boldsymbol{f}}^\ell)} \right) \right] \right] = \mathcal{F}_1 \left( \widehat{f}_1^\ell, \cdots, \widehat{f}_M^\ell \right) \\ \vdots \\ \widehat{f}_s^{\ell+1} = \nu \left[ \log \left( a_s e^{-\frac{\zeta}{\tau}} \right) - \log \left[ \sum_{j=1}^N \left( \frac{b_j e^{-C_{sj}/\epsilon}}{\mathscr{W}_{\epsilon,j}(\widehat{\boldsymbol{f}}^\ell)} \right) \right] \right] = \mathcal{F}_s \left( \widehat{f}_1^\ell, \cdots, \widehat{f}_M^\ell \right) \\ \vdots \\ \widehat{f}_M^{\ell+1} = \nu \left[ \log \left( a_M e^{-\frac{\zeta}{\tau}} \right) - \log \left[ \sum_{j=1}^N \left( \frac{b_j e^{-C_{Mj}/\epsilon}}{\mathscr{W}_{\epsilon,j}(\widehat{\boldsymbol{f}}^\ell)} \right) \right] \right] = \mathcal{F}_M \left( \widehat{f}_1^\ell, \cdots, \widehat{f}_M^\ell \right), \end{cases} \tag{33}$$

By taking the gradient on $\mathcal{F}_s(\widehat{f}_s^\ell)$ w.r.t $\widehat{f}_s^\ell$, we can observe that:

$$\frac{\partial \mathcal{F}_s(\widehat{f}_s^\ell)}{\partial \widehat{f}_s^\ell} = -\frac{\tau \epsilon}{\tau + \epsilon} \frac{1}{\sum_{j=1}^N \left[ \frac{\exp\left(-\frac{C_{sj}}{\epsilon}\right)}{\mathscr{W}_{\epsilon,j}(\widehat{\boldsymbol{f}}^\ell)} \right] b_j} \frac{\partial}{\partial \widehat{f}_s^\ell} \left( \sum_{j=1}^N \left[ \frac{\exp\left(-\frac{C_{sj}}{\epsilon}\right)}{\mathscr{W}_{\epsilon,j}(\widehat{\boldsymbol{f}}^\ell)} \right] b_j \right)$$

$$= \frac{\tau}{\tau + \epsilon} \frac{1}{\sum_{j=1}^N \left[ \frac{\exp\left(-\frac{C_{sj}}{\epsilon}\right)}{\mathscr{W}_{\epsilon,j}(\widehat{\boldsymbol{f}}^\ell)} \right] b_j} \underbrace{\sum_{j=1}^N \left[ \frac{b_j \exp\left(-\frac{C_{sj}}{\epsilon}\right)}{\mathscr{W}_{\epsilon,j}(\widehat{\boldsymbol{f}}^\ell)} \cdot \frac{\exp\left(\frac{\widehat{f}_s^\ell - C_{sj}}{\epsilon}\right)}{\mathscr{W}_{\epsilon,j}(\widehat{\boldsymbol{f}}^\ell)} \right]}_{<1} \tag{34}$$

$$< 1$$

Likewise we can obtain the result:

$$\mathscr{F}_s \left( \widehat{f}_1^\ell, \cdots, \widehat{f}_M^\ell \right) = \left| \frac{\partial \mathcal{F}_s(\widehat{f}_1^\ell)}{\partial \widehat{f}_1^\ell} \right| + \cdots + \left| \frac{\partial \mathcal{F}_s(\widehat{f}_s^\ell)}{\partial \widehat{f}_s^\ell} \right| + \cdots + \left| \frac{\partial \mathcal{F}_s(\widehat{f}_M^\ell)}{\partial \widehat{f}_M^\ell} \right|$$

$$= \frac{\tau}{\tau + \epsilon} \frac{1}{\sum_{j=1}^N \left[ \frac{\exp\left(-\frac{C_{sj}}{\epsilon}\right)}{\mathscr{W}_{\epsilon,j}(\widehat{\boldsymbol{f}}^\ell)} \right] b_j} \sum_{j=1}^N \left[ \frac{b_j \exp\left(-\frac{C_{sj}}{\epsilon}\right)}{\mathscr{W}_{\epsilon,j}(\widehat{\boldsymbol{f}}^\ell)} \cdot \sum_{u=1}^M \left( \frac{\exp\left(\frac{\widehat{f}_u^\ell - C_{uj}}{\epsilon}\right)}{\mathscr{W}_{\epsilon,j}(\widehat{\boldsymbol{f}}^\ell)} \right) \right] \tag{35}$$

$$= \frac{\tau}{\tau + \epsilon} < 1$$

We can easily conclude that:

$$
\begin{cases}
\mathscr{F}_1\left(\widehat{f}_1^\ell, \cdots, \widehat{f}_M^\ell\right) < 1 \\
\vdots \\
\mathscr{F}_s\left(\widehat{f}_1^\ell, \cdots, \widehat{f}_M^\ell\right) < 1 \\
\vdots \\
\mathscr{F}_M\left(\widehat{f}_1^\ell, \cdots, \widehat{f}_M^\ell\right) < 1
\end{cases}
\tag{36}
$$

Therefore, we can conclude that the proposed method guarantees convergence according to Theorem 2.9 in (Mathews, 2004). $\qquad\square$

## C. Algorithm for ETM-Based Method on SemiUOT

We also provide the pseudo algorithm of the proposed ETM-Based approachs (e.g., ETM-Exact, ETM-Approx and ETM-Refine) for solving SemiUOT in Alg.1 to make a more clear illustration.

---

**Algorithm 1** The algorithm of ETM-Based method on SemiUOT

---

**Input:** $C$: cost matrix; $a, b$: initial marginal probability; $\tau, \epsilon$: Hyper parameters.

    Randomly initialize the value of $f^{\text{init}}$.

    Choose ETM-Exact, ETM-Approx or ETM-Refine on SemiUOT for optimization.

**(1) Function:** ETM-Exact on SemiUOT($C, a, b, \tau, f^{t=0} = f^{\text{init}}$)

    Optimize $f$ via L-BFGS algorithm on $L_{\text{P}}$ as:

$$
\min_{f} L_{\text{P}} = \tau \sum_{i=1}^{M} a_i e^{-\frac{f_i + \zeta}{\tau}} - \sum_{j=1}^{N} \left[ \inf_{k \in [M]} [C_{kj} - f_k] - \zeta \right] b_j,
$$

    Optimize $g$ via $g_j = \inf_{k \in [M]} (C_{kj} - f_k^t)$.

    Optimize $\zeta$ via $\zeta = \tau[\log(\sum_{i=1}^{M} a_i \exp(-f_i/\tau)) - \log(\sum_{j=1}^{N} b_j)]$ as shown in Eq.(26).

**Return**: The optimal solutions of $f^*$, $g^*$ and $\zeta^*$.

**(2) Function:** ETM-Approx on SemiUOT($C, a, b, \tau, \widehat{f}^{t=0} = f^{\text{init}}$)

    Optimize $\widehat{f}$ via Proposition 2 on $\widehat{L}_{\text{P}}$ as:

$$
\min_{\widehat{f}} \widehat{L}_{\text{P}} = \tau \sum_{i=1}^{M} a_i e^{-\frac{\widehat{f}_i + \zeta}{\tau}} + \sum_{j=1}^{N} b_j \left[ \epsilon \log \left[ \sum_{k=1}^{M} e^{\frac{\widehat{f}_k - C_{kj}}{\epsilon}} \right] + \zeta \right]
$$

    Optimize $\widehat{g}$ via $\widehat{g}_j = -\epsilon \log[\sum_{k=1}^{M} \exp((\widehat{f}_k - C_{kj})/\epsilon)]$.

    Optimize $\zeta$ via $\zeta = \tau[\log(\sum_{i=1}^{M} a_i \exp(-\widehat{f}_i/\tau)) - \log(\sum_{j=1}^{N} b_j)]$ as shown in Eq.(26).

**Return**: The optimal solutions of $\widehat{f}^*$, $\widehat{g}^*$ and $\zeta^*$.

**(3) Function:** ETM-Refine on SemiUOT($C, a, b, \tau, \widehat{f}^{t=0} = f^{\text{init}}$)

    Obtain $\widehat{f}^* = $ ETM-Approx on SemiUOT($C, a, b, \tau, \widehat{f}^{t=0} = f^{\text{init}}$).

    Obtain $f^* = $ ETM-Exact on SemiUOT($C, a, b, \tau, f^{t=0} = \widehat{f}^*$).

**Return**: The optimal solutions of $f^*$, $g^*$ and $\zeta^*$.

---

## D. Proof of Proposition 3

**Proposition 3.** (Principles of Equivalent Transformation Mechanism for UOT) *Given UOT with KL-Divergence $\mathcal{J}_{\mathrm{UOT}}$, one can obtain its Fenchel-Lagrange multipliers form as below:*

$$\min_{\boldsymbol{u},\boldsymbol{v},\zeta} \left[ \tau_a \sum_{i=1}^{M} a_i e^{-\frac{u_i+\zeta}{\tau_a}} + \tau_b \sum_{j=1}^{N} b_j e^{-\frac{v_j-\zeta}{\tau_b}} \right] \tag{37}$$

$$s.t.\ u_i + v_j + s_{ij} = C_{ij}, \quad s_{ij} \ge 0.$$

*where $\boldsymbol{u}$, $\boldsymbol{v}$, $\boldsymbol{s}$ and $\zeta$ denotes Lagrange multipliers. Moreover, UOT problem can also be transformed into classic optimal transport as follows:*

$$\min_{\boldsymbol{\pi} \ge 0} \mathcal{J}_{\mathrm{U}} = \langle \boldsymbol{C}, \boldsymbol{\pi} \rangle$$

$$s.t. \begin{cases} \boldsymbol{\pi}\mathbf{1}_N = \boldsymbol{a} \odot \exp\left(-\frac{\boldsymbol{u}^* + \zeta^*}{\tau_a}\right) = \boldsymbol{\alpha} \\ \boldsymbol{\pi}^\top \mathbf{1}_M = \boldsymbol{b} \odot \exp\left(-\frac{\boldsymbol{v}^* - \zeta^*}{\tau_b}\right) = \boldsymbol{\beta} \end{cases} \tag{38}$$

*Note that when $\tau_a, \tau_b \to \infty$, the source and target marginal probability can be determined as $\boldsymbol{\pi}\mathbf{1}_N = \sqrt{\omega}\boldsymbol{a}$ and $\boldsymbol{\pi}^\top \mathbf{1}_M = \boldsymbol{b}/\sqrt{\omega}$ where $\omega = \langle \boldsymbol{b}, \mathbf{1}_N \rangle / \langle \boldsymbol{a}, \mathbf{1}_M \rangle$ respectively.*

*Proof.* To start with, we first rewrite the optimization problem as below:

$$\min_{\boldsymbol{\pi} \ge 0} J = \langle \boldsymbol{C}, \boldsymbol{\pi} \rangle + \tau_a \mathrm{KL}\left(\boldsymbol{\pi}\mathbf{1}_N \| \boldsymbol{a}\right) + \tau_b \mathrm{KL}(\boldsymbol{\pi}^\top \mathbf{1}_M \| \boldsymbol{b})$$

$$s.t.\ (\text{Optional}):\ \boldsymbol{\pi}\mathbf{1}_N = \boldsymbol{\alpha}, \quad \boldsymbol{\pi}^\top \mathbf{1}_M = \boldsymbol{\beta} \tag{39}$$

where $\boldsymbol{\alpha}$ and $\boldsymbol{\beta}$ denote the marginal probabilities for source and target domains respectively. Note that we do not need the true value fo $\boldsymbol{\alpha}$ and $\boldsymbol{\beta}$ beforehand. That is, the constraints here are optional for the following UOT deduction. The Lagrange multipliers of UOT with KL-Divergence is given as:

$$\max_{\boldsymbol{s} \ge 0, \boldsymbol{u}, \boldsymbol{v}, \zeta} \min_{\boldsymbol{\pi} \ge 0} \mathcal{J}_{\mathrm{UOT}} = \tau_a \mathrm{KL}\left(\boldsymbol{\pi}\mathbf{1}_N \| \boldsymbol{a}\right) + \langle \boldsymbol{u} + \zeta, \boldsymbol{\pi}\mathbf{1}_N \rangle + \tau_b \mathrm{KL}(\boldsymbol{\pi}^\top \mathbf{1}_M \| \boldsymbol{b}) + \langle \boldsymbol{v} - \zeta, \boldsymbol{\pi}^\top \mathbf{1}_M \rangle + \mathscr{C}_{\mathrm{UOT}} \tag{40}$$

where $\mathscr{C}_{\mathrm{UOT}} = \sum_{i,j}(C_{ij} - u_i - v_j - s_{ij})\pi_{ij} = \langle \boldsymbol{C} - \boldsymbol{u} \otimes \mathbf{1}_N^\top - \mathbf{1}_M \otimes \boldsymbol{v}^\top - \boldsymbol{s}, \boldsymbol{\pi} \rangle$ and $\boldsymbol{u}$, $\boldsymbol{v}$ and $\zeta$ are dual variables. By taking the differentiation on $\pi_{ij}$ we have:

$$\frac{\partial \mathcal{J}_{\mathrm{UOT}}}{\partial \pi_{ij}} = \left[ \tau_a \log \frac{\sum_{j=1}^{N} \pi_{ij}}{a_i} + u_i + \zeta \right] + \left[ \tau_b \log \frac{\sum_{i=1}^{M} \pi_{ij}}{b_j} + v_j - \zeta \right] + (C_{ij} - u_i - v_j - s_{ij})$$

$$= C_{ij} + \tau_a \log \frac{\sum_{j=1}^{N} \pi_{ij}}{a_i} + \tau_b \log \frac{\sum_{i=1}^{M} \pi_{ij}}{b_j} - s_{ij} = 0 \tag{41}$$

Then we can obtain the results:

$$\begin{cases} \sum_{j=1}^{N} \pi_{ij} = a_i \exp\left(-\frac{u_i + \zeta}{\tau_a}\right) \\ \sum_{i=1}^{M} \pi_{ij} = b_j \exp\left(-\frac{v_j - \zeta}{\tau_b}\right) \\ C_{ij} - u_i - v_j - s_{ij} = 0 \end{cases} \tag{42}$$

By taking the above results into KL-Divergence, we can further simplify the results:

$$\begin{cases} \tau_a \mathrm{KL}\left(\boldsymbol{\pi}\mathbf{1}_N \| \boldsymbol{a}\right) + \langle \boldsymbol{u} + \zeta, \boldsymbol{\pi}\mathbf{1}_N \rangle = \sum_{i=1}^{M} \left[ -\tau_a a_i \exp\left(-\frac{f_i + \zeta}{\tau_a}\right) + \tau_a a_i \right] \\ \tau_b \mathrm{KL}\left(\boldsymbol{\pi}^\top \mathbf{1}_M \| \boldsymbol{b}\right) + \langle \boldsymbol{v} - \zeta, \boldsymbol{\pi}^\top \mathbf{1}_M \rangle = \sum_{j=1}^{N} \left[ -\tau_b b_j \exp\left(-\frac{g_j - \zeta}{\tau_b}\right) + \tau_b b_j \right] \end{cases} \tag{43}$$

Therefore we can obtain its Fenchel-Lagrange multipliers form of UOT as:

$$\min_{\boldsymbol{u},\boldsymbol{v},\zeta} \mathcal{J}_{\text{UOT}} = -\tau_a \text{KL}\left(\boldsymbol{\pi}\mathbf{1}_N \| \boldsymbol{a}\right) - \langle \boldsymbol{u} + \zeta, \boldsymbol{\pi}\mathbf{1}_N \rangle - \tau_b \text{KL}(\boldsymbol{\pi}^\top \mathbf{1}_M \| \boldsymbol{b}) - \langle \boldsymbol{v} - \zeta, \boldsymbol{\pi}^\top \mathbf{1}_M \rangle$$

$$= \tau_a \exp\left(-\frac{\zeta}{\tau_a}\right) \left\langle \boldsymbol{a}, \exp\left(-\frac{\boldsymbol{u}}{\tau_a}\right)\right\rangle + \tau_b \exp\left(\frac{\zeta}{\tau_b}\right) \left\langle \boldsymbol{b}, \exp\left(-\frac{\boldsymbol{v}}{\tau_b}\right)\right\rangle + \mathcal{O}_{\text{Const}} \tag{44}$$

$$s.t.\ u_i + v_j \le C_{ij}$$

where $\mathcal{O}_{\text{Const}} = -\sum_{i=1}^{M} \tau_a a_i - \sum_{j=1}^{N} \tau_b b_j$ and we can neglect it during the following calculation. Once we obtain the optimal solution on $\boldsymbol{u}^*$, $\boldsymbol{v}^*$ and $\zeta^*$, the KL-Divergence will turn out to be constants and therefore the original optimization problem can be transformed into classic optimal transport. Finally we can obtain the optimal solution on $\zeta$ by considering $\frac{\partial \mathcal{J}_{\text{UOT}}}{\partial \zeta} = 0$ as below:

$$\zeta = \frac{\tau_a \tau_b}{\tau_a + \tau_b} \left[\log\left\langle \boldsymbol{a}, \exp\left(-\frac{\boldsymbol{u}}{\tau_a}\right)\right\rangle - \log\left\langle \boldsymbol{b}, \exp\left(-\frac{\boldsymbol{v}}{\tau_b}\right)\right\rangle\right]. \tag{45}$$

Once we set $\tau_a \to \infty$ and $\tau_b \to \infty$, the results of the limitation will be shown as:

$$\lim_{\tau_a \to +\infty, \tau_b \to +\infty} a_i \exp\left(-\frac{u_i + \zeta}{\tau_a}\right) = \lim_{\tau_a \to +\infty, \tau_b \to +\infty} a_i \exp\left(-\frac{\zeta}{\tau_a}\right) = a_i \sqrt{\frac{\langle \boldsymbol{b}, \mathbf{1}_N\rangle}{\langle \boldsymbol{a}, \mathbf{1}_M\rangle}} = \sqrt{\omega} a_i$$

$$\lim_{\tau_a \to +\infty, \tau_b \to +\infty} b_j \exp\left(-\frac{v_j - \zeta}{\tau_b}\right) = \lim_{\tau_a \to +\infty, \tau_b \to +\infty} b_j \exp\left(-\frac{\zeta}{\tau_b}\right) = b_j \sqrt{\frac{\langle \boldsymbol{a}, \mathbf{1}_M\rangle}{\langle \boldsymbol{b}, \mathbf{1}_N\rangle}} = \frac{1}{\sqrt{\omega}} b_j \tag{46}$$

Therefore we conclude the proof of the Proposition 3. □

## E. Illustrations of Optimization 1

**Optimization 1.** (Calculation of ETM-Approx approach for UOT) To start with, we first review the Exact UOT Equation is defined as:

$$\min_{\boldsymbol{u},\zeta} L_{\text{U}} = \tau_a \sum_{i=1}^{M} a_i \exp\left(-\frac{u_i + \zeta}{\tau_a}\right) + \tau_b \exp\left(\frac{\zeta}{\tau_b}\right) \sum_{j=1}^{N} b_j \exp\left(-\frac{v_j}{\tau_b}\right)$$

$$= \tau_a \sum_{i=1}^{M} a_i \exp\left(-\frac{u_i + \zeta}{\tau_a}\right) + \tau_b \exp\left(\frac{\zeta}{\tau_b}\right) \sum_{j=1}^{N} b_j \exp\left(\frac{\sup_{k \in [M]}\left(u_k - C_{kj}\right)}{\tau_b}\right) \tag{47}$$

where $v_j = -\sup_{k \in [M]}\left(u_k - C_{kj}\right)$ meanwhile the marginal probabilities are set as $\boldsymbol{\pi}\mathbf{1}_N = \boldsymbol{a} \odot \exp\left(-(\boldsymbol{u} + \zeta)/\tau_a\right) = \boldsymbol{\alpha}$ and $\boldsymbol{\pi}^\top \mathbf{1}_M = \boldsymbol{b} \odot \exp\left(-(\boldsymbol{v} - \zeta)/\tau_b\right) = \boldsymbol{\beta}$. Since the optimization problem in Eq.(8) is convex, we can also utilize block gradient descend to optimize the problem. Specifically, we first fix $v^l$ and optimize variable $u^l$ at the $l$-th iteration by replacing the original marginal probability $\boldsymbol{b}$ in Eq.(4) with $\boldsymbol{\beta}$ accordingly to transform UOT into SemiUOT problem:

$$\min_{\boldsymbol{\pi} \ge 0} J_{\text{U}}^u = \langle \boldsymbol{C}, \boldsymbol{\pi} \rangle + \tau_a \text{KL}\left(\boldsymbol{\pi}\mathbf{1}_N \| \boldsymbol{a}\right)$$

$$s.t.\ \begin{cases} (\text{Constraint}):\ \boldsymbol{\pi}^\top \mathbf{1}_M = \boldsymbol{b} \odot \exp\left(-\frac{\boldsymbol{v} - \zeta}{\tau_b}\right) = \boldsymbol{\beta} \\ (\text{Optional}):\ \boldsymbol{\pi}\mathbf{1}_N = \boldsymbol{a} \odot \exp\left(-\frac{\boldsymbol{u} + \zeta}{\tau_a}\right) = \boldsymbol{\alpha} \end{cases} \tag{48}$$

At that time, the Fenchel-Lagrange multipliers form of Eq.(48) is given via the Proposition 1:

$$\min_{\boldsymbol{u}} L_{\text{U}}^u = \tau_a \sum_{i=1}^{M} a_i \exp\left(-\frac{\widetilde{u}_i + \zeta}{\tau_a}\right) - \sum_{j=1}^{N} \beta_j(\widetilde{v}_j - \zeta)$$

$$= \tau_a \sum_{i=1}^{M} a_i \exp\left(-\frac{u_i + \zeta}{\tau_a}\right) - \sum_{j=1}^{N} \left(\inf_{k \in [M]}\left[C_{kj} - u_k\right] - \zeta\right)\beta_j \tag{49}$$

Note that $\widetilde{\boldsymbol{u}}$ and $\widetilde{\boldsymbol{v}}$ denote the Lagrange multiplier for Eq.(48) while we have $\widetilde{v}_j = \inf_{k \in [M]}\left[C_{kj} - u_k\right] = v_j$ and $\widetilde{\boldsymbol{u}} = \boldsymbol{u}$. To further accelerate the optimization process, we consider to make a smooth approximation on replacing $\inf(\cdot)$ as $\inf_{k \in [M]}[C_{kj} - u_k] \approx -\epsilon \log[\sum_{k=1}^{M} e^{\frac{u_k - C_{kj}}{\epsilon}}] = \widehat{v}_j$. Therefore, we first fix $\widehat{v}^l$ and optimize variable $\widehat{u}^l$ at the $l$-th iteration

to solve the following equation on $\widehat{L}_U^u$ accordingly:

$$\min_{\widehat{u}} \widehat{L}_U^u = \tau_a \sum_{i=1}^{M} a_i \exp\left(-\frac{\widehat{u}_i + \zeta}{\tau_a}\right) + \sum_{j=1}^{N} \beta_j \left[\epsilon \log\left[\sum_{k=1}^{M} e^{\frac{\widehat{u}_k - C_{kj}}{\epsilon}}\right] + \zeta\right]$$

$$= \tau_a \sum_{i=1}^{M} a_i \exp\left(-\frac{\widehat{u}_i + \zeta}{\tau_a}\right) + \sum_{j=1}^{N} b_j \exp\left(-\frac{\widehat{v}_j - \zeta}{\tau_b}\right) \left[\epsilon \log\left[\sum_{k=1}^{M} e^{\frac{\widehat{u}_k - C_{kj}}{\epsilon}}\right] + \zeta\right] \tag{50}$$

At that time we adopt fixed-point iteration method to optimize $\widehat{u}$ accordingly based on the Proposition 2:

$$\begin{cases} \widehat{u}_1^{\ell+1} = \frac{\tau_a \epsilon}{\tau_a + \epsilon} \left[\log\left(a_1 e^{-\frac{\zeta}{\tau_a}}\right) - \log\left[\sum_{j=1}^{N} \left(\frac{\beta_j e^{-C_{1j}/\epsilon}}{\mathscr{W}_{\epsilon,j}(\widehat{u}^\ell)}\right)\right]\right] = \mathcal{U}_1\left(\widehat{u}_1^\ell, \cdots, \widehat{u}_M^\ell\right) \\ \vdots \\ \widehat{u}_s^{\ell+1} = \frac{\tau_a \epsilon}{\tau_a + \epsilon} \left[\log\left(a_s e^{-\frac{\zeta}{\tau_a}}\right) - \log\left[\sum_{j=1}^{N} \left(\frac{\beta_j e^{-C_{sj}/\epsilon}}{\mathscr{W}_{\epsilon,j}(\widehat{u}^\ell)}\right)\right]\right] = \mathcal{U}_s\left(\widehat{u}_1^\ell, \cdots, \widehat{u}_M^\ell\right) \\ \vdots \\ \widehat{u}_M^{\ell+1} = \frac{\tau_a \epsilon}{\tau_a + \epsilon} \left[\log\left(a_M e^{-\frac{\zeta}{\tau_a}}\right) - \log\left[\sum_{j=1}^{N} \left(\frac{\beta_j e^{-C_{Mj}/\epsilon}}{\mathscr{W}_{\epsilon,j}(\widehat{u}^\ell)}\right)\right]\right] = \mathcal{U}_M\left(\widehat{u}_1^\ell, \cdots, \widehat{u}_M^\ell\right), \end{cases} \tag{51}$$

The iteration process can be shown to converge based on Proposition 2. After that we fix $\widehat{u}$ and optimize variable $\widehat{v}$ via $\widehat{v}_j = -\epsilon \log[\sum_{k=1}^{M} \exp((\widehat{u}_k - C_{kj})/\epsilon)]$. We can achieve the optimal solution on $\widehat{u}^*$ and $\widehat{v}^*$ via iteratively computing via the above procedure accordingly. Finally, we update $\zeta$ via $\zeta = (\tau_a \tau_b/(\tau_a + \tau_b))[\log(\sum_{i=1}^{M} a_i \exp(-\widehat{u}_i^*/\tau_a)) - \log(\sum_{j=1}^{N} b_j \exp(-\widehat{v}_j^*/\tau_b))]$.

## F. Algorithm for ETM-Based Method on UOT

We also provide the pseudo algorithm of the proposed ETM-Based approachs (e.g., ETM-Exact, ETM-Approx and ETM-Refine) for solving UOT in Alg.2 to make a more clear illustration.

## G. Proof of Proposition 4

**Proposition 4.** (The Definition and Usage of KKT-Multiplier Regularization) *Given any OT with multiplier $s$, one can obtain accurate solution $\pi^*$ via proposed KKT-multiplier regularization term $\mathcal{G}(\pi, s) = \langle \pi, s \rangle$, which formulates Multiplier Regularized Optimal Transport (MROT):*

$$\min_{\pi \geq 0} \mathcal{J}_G = \langle C, \pi \rangle + \eta_G \langle \pi, s \rangle + \eta_{\text{Reg}} \mathcal{L}_{\text{Reg}}(\pi)$$

$$s.t. \ \pi \mathbf{1}_N = \alpha, \quad \pi^\top \mathbf{1}_M = \beta, \tag{52}$$

*where $\mathcal{L}_{\text{Reg}}(\pi)$ denotes the regularization term on $\pi$. $\alpha$, $\beta$ denote the final marginal probabilities obtained by ETM-based approach and $\eta_{\text{Reg}}$, $\eta_G$ denotes the hyper parameter. Ideally, $\eta_G$ should be set as a relatively large number. Meanwhile the dual form of MROT is given as:*

$$\max_{\psi, \phi} L_G = \langle \alpha, \psi \rangle + \langle \beta, \phi \rangle - \eta_{\text{Reg}} \mathcal{L}_{\text{Reg}}^* \left(\frac{\psi_i + \phi_j - \widetilde{C}_{ij}}{\eta_{\text{Reg}}}\right) \tag{53}$$

*where $\widetilde{C}_{ij} = C_{ij} + \eta_G s_{ij}$ and $\phi$ and $\psi$ denote the Lagrange multipliers for MROT. $\mathcal{L}_{\text{Reg}}^*(\cdot)$ denotes the conjugate function of $\mathcal{L}_{\text{Reg}}(\cdot)$ and one can figure out the matching results of $\pi$ via solving $\nabla_{\pi_{ij}} \mathcal{L}_{\text{Reg}}(\pi_{ij}) = (\psi_i + \phi_j - \widetilde{C}_{ij})/\eta_{\text{Reg}}$.*

---

**Algorithm 2** The algorithm of ETM-Based method on UOT

---

**Input:** $C$: cost matrix; $a, b$: initial marginal probability; $\tau_a, \tau_b, \epsilon$: Hyper parameters.

Randomly initialize the value of $u^{\text{init}}$.

Choose ETM-Exact, ETM-Approx or ETM-Refine on UOT for optimization.

**(1) Function:** ETM-Exact on UOT($C, a, b, \tau_a, \tau_b, u^{t=0} = u^{\text{init}}$)

Optimize $u$ L-BFGS algorithm to optimize $L_{\text{U}}$ as:

$$\min_{u} L_{\text{U}} = \tau_a \sum_{i=1}^{M} a_i e^{-\frac{u_i + \zeta}{\tau_a}} + \tau_b e^{\frac{\zeta}{\tau_b}} \sum_{j=1}^{N} b_j e^{\frac{\sup_{k \in [M]} \left( u_k - C_{kj} \right)}{\tau_b}},$$

Optimize $v$ via $v_j = \inf_{k \in [M]} (C_{kj} - u_k)$.

Optimize $\zeta$ via $\zeta = \frac{\tau_a \tau_b}{\tau_a + \tau_b} \left[ \log \left\langle a, \exp \left( -\frac{u}{\tau_a} \right) \right\rangle - \log \left\langle b, \exp \left( -\frac{v}{\tau_b} \right) \right\rangle \right]$ as shown in Eq.(45).

**Return**: The optimal solutions of $u^*, v^*$ and $\zeta^*$.

**(2) Function:** ETM-Approx on UOT($C, a, b, \tau_a, \tau_b, \widehat{u}^{t=0} = u^{\text{init}}$)

Randomly initialize the value of $\widehat{v}^{t'=1}$.

**for** $t' = 1$ to $T'$ **do**

Optimize $\widehat{u}^{t'}$ via Proposition 2 to optimize $\widehat{L}_{\text{U}}^u$ as:

$$\min_{\widehat{u}} \widehat{L}_{\text{U}}^u = \tau_a \sum_{i=1}^{M} a_i \exp \left( -\frac{\widehat{u}_i + \zeta}{\tau_a} \right) + \sum_{j=1}^{N} b_j \exp \left( -\frac{\widehat{v}_j - \zeta}{\tau_b} \right) \left[ \epsilon \log \left[ \sum_{k=1}^{M} e^{\frac{\widehat{u}_k - C_{kj}}{\epsilon}} \right] + \zeta \right]$$

Optimize $\widehat{v}^{t'}$ via $\widehat{v}_j^{t'} = -\epsilon \log[\sum_{k=1}^{M} \exp((\widehat{u}_k^{t'} - C_{kj})/\epsilon)]$.

**end for**

Optimize $\zeta$ via $\zeta = \frac{\tau_a \tau_b}{\tau_a + \tau_b} \left[ \log \left\langle a, \exp \left( -\frac{\widehat{u}}{\tau_a} \right) \right\rangle - \log \left\langle b, \exp \left( -\frac{\widehat{v}}{\tau_b} \right) \right\rangle \right]$ as shown in Eq.(45).

**Return**: The optimal solutions of $\widehat{u}^*, \widehat{v}^*$ and $\zeta^*$.

**(3) Function:** ETM-Refine on UOT($C, a, b, \tau_a, \tau_b, \widehat{u}^{t=0} = u^{\text{init}}$)

Obtain $\widehat{u}^* = $ ETM-Approx on UOT($C, a, b, \tau_a, \tau_b, \widehat{u}^{t=0} = u^{\text{init}}$).

Obtain $u^* = $ ETM-Exact on UOT($C, a, b, \tau_a, \tau_b, u^{t=0} = \widehat{u}^*$).

**Return**: The optimal solutions of $u^*, v^*$ and $\zeta^*$.

---

*Proof.* We first provide the Lagrange multiplier of MROT as:

$$\max_{\psi, \phi} \min_{\pi \geq 0} \mathcal{J}_{\text{MROT}} = \langle C, \pi \rangle + \eta_G \langle \pi, s \rangle + \eta_{\text{Reg}} \mathcal{L}_{\text{Reg}}(\pi) - \langle \psi, \pi \mathbf{1}_N - \alpha \rangle - \langle \phi, \pi^\top \mathbf{1}_M - \beta \rangle$$

$$= \langle \alpha, \psi \rangle + \langle \beta, \phi \rangle + \eta_{\text{Reg}} \inf_{\pi} \left[ \sum_{i,j} \left[ \frac{C_{ij} + \eta_G s_{ij} - f_i - g_j}{\eta_{\text{Reg}}} \pi_{ij} + \mathcal{L}_{\text{Reg}}(\pi_{ij}) \right] \right]$$

$$= \langle \alpha, \psi \rangle + \langle \beta, \phi \rangle - \eta_{\text{Reg}} \sup_{\pi} \left[ \sum_{i,j} \left[ \frac{f_i + g_j - \widetilde{C}_{ij}}{\eta_{\text{Reg}}} \pi_{ij} - \mathcal{L}_{\text{Reg}}(\pi_{ij}) \right] \right] \tag{54}$$

$$= \langle \alpha, \psi \rangle + \langle \beta, \phi \rangle - \eta_{\text{Reg}} \mathcal{L}_{\text{Reg}}^* \left( \frac{\psi_i + \phi_j - \widetilde{C}_{ij}}{\eta_{\text{Reg}}} \right)$$

At that time we have the following results:

$$\begin{cases} \dfrac{\partial \mathcal{J}_{\text{MROT}}}{\partial \psi_i} = 0 \\ \dfrac{\partial \mathcal{J}_{\text{MROT}}}{\partial \phi_j} = 0 \end{cases} \Rightarrow \begin{cases} \nabla_{\psi_i} \mathcal{L}_{\text{Reg}}^* \left( \dfrac{\psi_i + \phi_j - \widetilde{C}_{ij}}{\eta_{\text{Reg}}} \right) = \alpha_i \\ \nabla_{\phi_j} \mathcal{L}_{\text{Reg}}^* \left( \dfrac{\psi_i + \phi_j - \widetilde{C}_{ij}}{\eta_{\text{Reg}}} \right) = \beta_j \end{cases} \tag{55}$$

By taking the differentiation on $\pi_{ij}$ we have:

$$\frac{\partial \mathcal{J}_{\text{MROT}}}{\partial \pi_{ij}} = \widetilde{C}_{ij} + \eta_{\text{Reg}} \nabla_{\pi_{ij}} \mathcal{L}_{\text{Reg}}(\pi_{ij}) - \psi_i - \phi_j = 0 \tag{56}$$

For instance, when $\mathcal{L}_{\text{Reg}}(\boldsymbol{\pi}) = -\langle \boldsymbol{\pi}, \log(\boldsymbol{\pi}) - 1 \rangle$ denotes as the entropy regularization term, the dual form of MROT-Ent is shown as:

$$
\begin{cases}
\displaystyle \max_{\boldsymbol{\psi},\boldsymbol{\phi}} \mathcal{J}_{\text{MROT}-\text{Ent}} = \langle \boldsymbol{\alpha}, \boldsymbol{\psi} \rangle + \langle \boldsymbol{\beta}, \boldsymbol{\phi} \rangle - \eta_{\text{Reg}} \sum_{i,j} \exp\left( \frac{\psi_i + \phi_j - \widetilde{C}_{ij}}{\eta_{\text{Reg}}} \right) \\[4mm]
\displaystyle \pi_{ij} = \exp\left( \frac{\psi_i + \phi_j - \widetilde{C}_{ij}}{\eta_{\text{Reg}}} \right)
\end{cases}
\tag{57}
$$

When $\mathcal{L}_{\text{Reg}}(\boldsymbol{\pi}) = \langle \boldsymbol{\pi}, \boldsymbol{\pi} \rangle / 2$ denotes as the square-norm regularization term, the dual form of MROT-Norm is shown as:

$$
\begin{cases}
\displaystyle \max_{\boldsymbol{\psi},\boldsymbol{\phi}} \mathcal{J}_{\text{MROT}-\text{Norm}} = \langle \boldsymbol{\alpha}, \boldsymbol{\psi} \rangle + \langle \boldsymbol{\beta}, \boldsymbol{\phi} \rangle - \frac{\eta_{\text{Reg}}}{2} \sum_{i,j} \left[ \frac{\psi_i + \phi_j - \widetilde{C}_{ij}}{\eta_{\text{Reg}}} \right]_+^2 \\[4mm]
\displaystyle \pi_{ij} = \left[ \frac{\psi_i + \phi_j - \widetilde{C}_{ij}}{\eta_{\text{Reg}}} \right]_+
\end{cases}
\tag{58}
$$

Therefore we conclude the proof of the Proposition 4. $\qquad\square$

## H. Datasets on Domain Adaptations

**Datasets.** We conduct the unsupervised domain adaptation tasks on *Digits*, *Office-Home*, and *VisDA*. *Digits* is the classical dataset for digit classification which contains three standard digit classification datasets: **MNIST** (Lecun et al., 1998), **USPS** (Hull, 2002) and **SVHN** (Netzer et al., 2011). Each dataset consists of 10 classes of digits, ranging from 0 to 9. *Office-Home* (Venkateswara et al., 2017) is a standard benchmark dataset which includes 15,500 images in 65 object classes in office and home settings, forming four dissimilar domains: Artistic images (**Ar**), Clip Art (**Cl**), Product images (**Pr**) and Real-World (**Rw**). *VisDA* (Peng et al., 2018) is the large-scale cross-domain dataset in computer vision on two domains, i.e., **Synthetic** and **Real** with 280K images in 12 classes.

## I. Experiments on Partial Domain Adaptations

**Datasets.** We further conduct the domain adaptation tasks on new datasets, i.e., *Office-31* (Saenko et al., 2010) and *ImageCLEF* (Caputo et al., 2014). **Office-31** is the commonly-used computer vision dataset for domain adaptation with 4,652 images from three different domains: *Amazon* (**A**), *Webcam* (**W**) and *DSLR* (**D**). Target domain has the first 10 classes (alphabetical order) following (Cao et al., 2018). **ImageCLEF** contains 3 domains with 12 classes, i.e., *Caltech* (**C**), *ImageNet* (**I**) and *Pascal* (**P**). Target domain has the first 6 classes (alphabetical order) following (Luo et al., 2020).

**Baselines.** We involve **DeepJDOT** (Damodaran et al., 2018), **ROT** (Balaji et al., 2020), **JUMBOT** (Fatras et al., 2021), **ETN** (Cao et al., 2019), **AR** (Gu et al., 2021), **m-POT** (Nguyen et al., 2022), **MOT** (Luo & Ren, 2023), **DMP** (Luo et al., 2020) as the model baselines for the domain adaptation task.

- **DeepJDOT** (Damodaran et al., 2018) first adopts optimal transport into solving domain adaptation problem with deep learning framework.

- **ROT** (Balaji et al., 2020) adopts robust optimal transport into adversarial training for domain adaptation.

- **JUMBOT** (Fatras et al., 2021) adopts mini-batch unbalanced optimal transport method for domain adaptation.

- **ETN** (Cao et al., 2019) utilizes example transfer network to jointly learn domain-invariant representations and the progressive weighting scheme.

- **AR** (Gu et al., 2021) adopts adversarial reweighting strategy on source domain data for alignment.

- **m-POT** (Nguyen et al., 2022) adopts partial optimal transport method in the mini-batch settings for domain adaptation.

- **DMP** (Luo et al., 2020) adopt discriminative manifold propagation for domain adaptation.

- **MOT** (Luo & Ren, 2023) adopts masked unbalanced optimal transport technique on considering label information for PDA tasks.

*Table 3.* Classification accuracy (%) on *Office-31* for partial unsupervised domain adaptation

| Method | A→W | D→W | W→D | A→D | D→A | W→A | Avg |
|---|---|---|---|---|---|---|---|
| ResNet (He et al., 2016) | 75.6 | 96.3 | 98.1 | 83.4 | 83.9 | 85.0 | 87.1 |
| ETN (Cao et al., 2019) | 84.7 | 97.4 | 99.2 | 91.3 | 90.2 | 92.8 | 92.6 |
| JUMBOT (Fatras et al., 2021) | 90.2 | 98.9 | 99.3 | 94.5 | 93.8 | 93.4 | 95.0 |
| AR (Gu et al., 2021) | 93.5 | **100.0** | 99.7 | 96.8 | 95.5 | 96.0 | 96.9 |
| m-POT (Nguyen et al., 2022) | 96.2 | 99.5 | **100.0** | 97.6 | 94.4 | 95.3 | 97.2 |
| MOT (Luo & Ren, 2023) | 99.3 | **100.0** | **100.0** | 98.7 | 96.1 | 96.4 | 98.4 |
| MOT + UOT(ETM + MROT-Ent) | 99.4 | **100.0** | **100.0** | 98.9 | 96.8 | 97.3 | 98.7 |
| MOT + UOT(ETM + MROT-Norm) | 99.6 | **100.0** | **100.0** | 99.2 | 97.3 | 97.7 | 99.0 |
| MOT + SemiUOT(ETM + MROT-Ent) | **99.8** | **100.0** | **100.0** | 99.4 | 97.8 | 98.4 | 99.2 |
| MOT + SemiUOT(ETM + MROT-Norm) | 99.7 | **100.0** | **100.0** | **99.7** | **98.4** | **98.6** | **99.4** |

*Table 4.* Classification accuracy (%) on *ImageCLEF* for partial unsupervised domain adaptation

| Method | I→P | P→I | I→C | C→I | C→P | P→C | Avg |
|---|---|---|---|---|---|---|---|
| ResNet (He et al., 2016) | 78.3 | 86.9 | 91.0 | 84.3 | 72.5 | 91.5 | 84.1 |
| ETN (Cao et al., 2019) | 79.6 | 88.5 | 92.9 | 87.2 | 74.1 | 93.4 | 86.0 |
| JUMBOT (Fatras et al., 2021) | 80.1 | 91.3 | 93.6 | 90.9 | 75.7 | 94.2 | 87.6 |
| AR (Gu et al., 2021) | 83.1 | 92.8 | 94.5 | 92.4 | 76.3 | 95.0 | 89.0 |
| DMP (Luo et al., 2020) | 82.4 | 94.5 | 96.7 | 94.3 | 78.7 | 96.4 | 90.5 |
| MOT (Luo & Ren, 2023) | 87.7 | 95.0 | 98.0 | 95.0 | 87.0 | 98.7 | 93.6 |
| MOT + UOT(ETM + MROT-Ent) | 88.3 | 95.6 | 98.4 | 95.3 | 87.6 | 99.0 | 94.0 |
| MOT + UOT(ETM + MROT-Norm) | 88.7 | 95.9 | 98.7 | 95.8 | 88.0 | 99.1 | 94.4 |
| MOT + SemiUOT(ETM + MROT-Ent) | 89.1 | 96.2 | 99.2 | 96.1 | 88.5 | 99.4 | 94.8 |
| MOT + SemiUOT(ETM + MROT-Norm) | **89.6** | **96.7** | **99.4** | **96.5** | **89.1** | **99.6** | **95.2** |

**Performance.** We also conduct the partial domain adaptation tasks on Office-31 and ImageCLEF and the results are shown in Table.4-5. We can observe that the proposed ETM-Refine with MROT-Norm on SemiUOT achieves state-of-the-art performance on Office-31 and ImageCLEF.

## J. Experiments on Universal Domain Adaptations

We further conduct the experiments on universal domain adaptations. That is, there are shared labels between the source and target domains. Additionally, there are private labels specific to each domain (Farahani et al., 2021; Zhang & Gao, 2022). We conduct the universal domain adaptations on both *Office-31* and *Office-Home*. Specifically, we set the first 10 classes in alphabetical order as the common label set, the next 10 classes as source private label and the rest 11 classes as target private label for Office-31. Likewise, we set the first 10 classes in alphabetical order as the common label set, the next 5 classes as source private label and the rest 55 classes as target private label for Office-Home. We involve the following models as baselines: (1) **OSBP** (Saito et al., 2018) adopts domain adversarial learning for open-set domain adaptation, (2) **UAN** (You et al., 2019) utilizes transferability criterion for universal domain adaptation, (3) **CMU** (Fu et al., 2020) learns to detect open classes with uncertainty estimation, (4) **DCC** (Li et al., 2021) adopts domain consensus clustering for adaptation, (5) **TNT** (Chen et al., 2022) adopts evidential neighborhood contrastive learning for adaptation, (6) **UniOT** (Chang et al., 2022) adopts unbalanced optimal transport with adaptive filtering for transferring.

We adopt the same experimental settings as UniOT (Chang et al., 2022). We utilize the commonly-used H-score (Fu et al., 2020) to validate the final results as shown in Table.6-7. Note that UniOT + UOT(ETM + MROT) only replaces the entropic UOT in UniOT with our proposed ETM-Refine method with MROT. From that we can observe that UniOT + UOT(ETM + MROT) reaches the best performance, indicating that UOT with ETM + MROT can provide more accurate matching results. Moreover, we adopt T-SNE method (Van der Maaten & Hinton, 2008) to plot the source and target data features in the latent space as shown in Fig.8(a)-(b). We can find that: (1) original UniOT could lead to rather scattered features in the latent space. That is because UniOT with entropic UOT could lead to dense and inaccurate matching solutions which limits the model potentials. (2) Our proposed UniOT + UOT(ETM + MROT-Norm) can provide more compact features since it provides more accurate solutions. Thus it further illustrates the efficacy of our proposed ETM + MROT-Norm method.

*Table 5.* H-score (%) on *Office-Home* for universal unsupervised domain adaptation

| Method | Ar→Cl | Ar→Pr | Ar→Rw | Cl→Ar | Cl→Pr | Cl→Rw | Pr→Ar | Pr→Cl | Pr→Rw | Rw→Ar | Rw→Cl | Rw→Pr | Avg |
|---|---|---|---|---|---|---|---|---|---|---|---|---|---|
| ResNet (He et al., 2016) | 44.65 | 48.04 | 50.13 | 46.64 | 46.91 | 48.96 | 47.47 | 43.17 | 50.23 | 48.45 | 44.76 | 48.43 | 47.32 |
| OSBP (Saito et al., 2018) | 39.59 | 45.09 | 46.17 | 45.70 | 45.24 | 46.75 | 45.26 | 40.54 | 45.08 | 45.75 | 41.64 | 46.90 | 44.48 |
| UAN (You et al., 2019) | 51.64 | 51.70 | 54.30 | 61.74 | 57.63 | 61.86 | 50.38 | 47.62 | 61.46 | 62.87 | 52.61 | 65.19 | 56.58 |
| CMU (Fu et al., 2020) | 56.02 | 56.93 | 59.15 | 66.95 | 64.27 | 67.82 | 54.72 | 51.09 | 66.39 | 68.24 | 57.89 | 69.73 | 61.60 |
| DCC (Li et al., 2021) | 57.97 | 54.05 | 58.01 | 74.64 | 70.62 | 77.52 | 64.34 | 73.60 | 74.94 | 80.96 | **75.12** | 80.38 | 70.18 |
| TNT (Chen et al., 2022) | 61.90 | 74.60 | 80.20 | 73.50 | 71.40 | 79.60 | 74.20 | **69.50** | 82.70 | 77.30 | 70.10 | 81.20 | 74.70 |
| UniOT (Chang et al., 2022) | 67.27 | 80.54 | 86.03 | 73.51 | 77.33 | 84.28 | 75.54 | 63.33 | 85.99 | 77.77 | 65.37 | 81.92 | 76.57 |
| UniOT + UOT(ETM + MROT-Ent) | 68.63 | 81.72 | 87.94 | 75.88 | 79.03 | 86.21 | 77.29 | 68.77 | 87.14 | 78.59 | 73.62 | 82.83 | 78.97 |
| UniOT + UOT(ETM + MROT-Norm) | **69.02** | **81.95** | **88.36** | **76.12** | **79.36** | **86.49** | **77.03** | 69.25 | **87.30** | **78.93** | 74.18 | **82.96** | **79.25** |

*Table 6.* H-Score (%) on *Office-31* for universal unsupervised domain adaptation

| Method | A→D | A→W | D→A | D→W | W→A | W→D | Avg |
|---|---|---|---|---|---|---|---|
| ResNet (He et al., 2016) | 49.78 | 47.92 | 48.48 | 54.94 | 48.96 | 55.60 | 50.94 |
| OSBP (Saito et al., 2018) | 51.14 | 50.23 | 49.75 | 55.53 | 50.16 | 57.20 | 52.34 |
| UAN (You et al., 2019) | 59.68 | 58.61 | 60.11 | 70.62 | 60.34 | 71.42 | 63.46 |
| CMU (Fu et al., 2020) | 68.11 | 67.33 | 71.42 | 79.32 | 72.23 | 80.42 | 73.14 |
| DCC (Li et al., 2021) | 88.50 | 78.54 | 70.18 | 79.29 | 75.87 | 88.58 | 80.16 |
| TNT (Chen et al., 2022) | 85.70 | 80.40 | 83.80 | 92.00 | 79.10 | 91.20 | 85.37 |
| UniOT (Chang et al., 2022) | 86.97 | 88.48 | 88.35 | 98.83 | 87.60 | 96.57 | 91.13 |
| UniOT + UOT(ETM + MROT-Ent) | 88.25 | 89.62 | 89.47 | 99.48 | 89.10 | 97.94 | 92.31 |
| UniOT + UOT(ETM + MROT-Norm) | **88.67** | **90.14** | **90.03** | **99.58** | **89.42** | **98.46** | **92.72** |

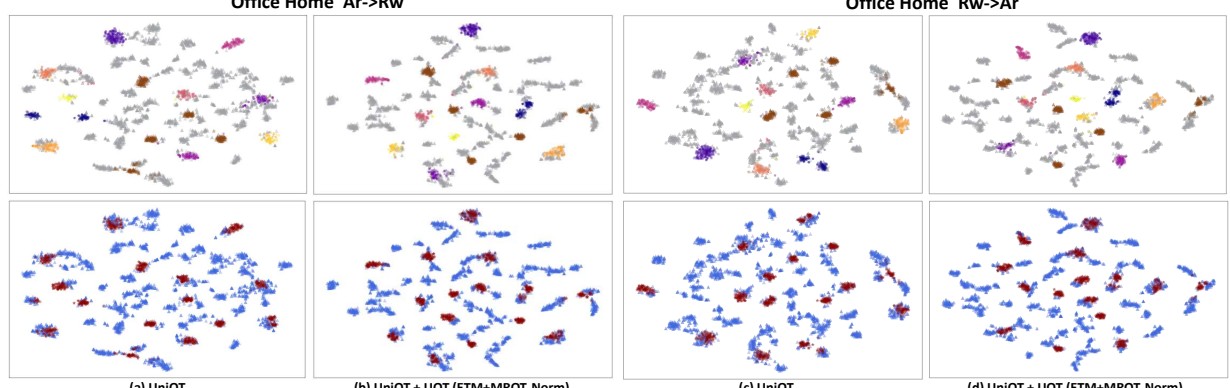

**Office Home Ar->Rw**     **Office Home Rw->Ar**

**(a) UniOT**  **(b) UniOT + UOT (ETM+MROT-Norm)**  **(c) UniOT**  **(d) UniOT + UOT (ETM+MROT-Norm)**

*Figure 5.* The T-SNE of data features on Ar → Rw (Office-Home) and Rw → Ar (Office-Home). The first row shows the original data sample distribution: The brown and gray colors denote the source and target private classes respectively. The rest are common label set with different colors. The second row indicates the mapping between source and target domain: The red and blue points denote the source and target samples respectively.

*Table 7.* Experimental results on Treatment Effect Estimation tasks.

| | ACIC (PEHE) | | ACIC (AUUC) | | IHDP (PEHE) | | IHDP (AUUC) | |
|---|---|---|---|---|---|---|---|---|
| | In-Sample | Out-Sample | In-Sample | Out-Sample | In-Sample | Out-Sample | In-Sample | Out-Sample |
| OLS (Angrist & Imbens, 1995) | 3.749 | 4.340 | 0.843 | 0.496 | 3.856 | 5.674 | 0.652 | 0.492 |
| TARNet (Shalit et al., 2017) | 3.236 | 3.254 | **0.886** | 0.662 | 0.749 | 1.788 | 0.654 | 0.711 |
| PSM (Rosenbaum & Rubin, 1983) | 5.228 | 5.094 | 0.884 | 0.745 | 3.219 | 4.634 | 0.740 | 0.681 |
| CFR-WASS (Shalit et al., 2017) | 3.128 | 3.207 | 0.873 | 0.669 | 0.657 | 1.704 | 0.656 | 0.715 |
| ESCFR (Wang et al., 2023) | 2.252 | 2.316 | 0.796 | 0.754 | 0.502 | 1.282 | 0.665 | 0.719 |
| ESCFR + UOT(ETM + MROT-Ent) | 2.327 | 2.261 | 0.839 | 0.814 | 0.497 | 1.275 | 0.769 | 0.763 |
| ESCFR + UOT(ETM + MROT-Norm) | **2.104** | **2.216** | 0.883 | **0.839** | **0.475** | **1.146** | **0.798** | **0.802** |

## K. Experiments on Treatment Effect Estimation

**Datasets for Treatment Effect Estimation.** We further conduct ETM on treatment effect estimation with two semi-synthetic datasets IHDP (Shalit et al., 2017) and ACIC (Yao et al., 2018). IHDP is set to estimate the effect of specialist home visits on infants' potential cognitive scores and it contains 747 observations and 25 covariates. ACIC includes 4802 observations and 58 covariates which comes from the collaborative perinatal project.

**Results.** We involve the following models as baselines: (1) **OLS** (Angrist & Imbens, 1995) utilizes least square regression with treatment as covariates, (2) **TARNet** (Shalit et al., 2017) adopts integral orobability metrics for adaptation, (3) **PSM** (Rosenbaum & Rubin, 1983) adopts propensity score for causal effects, (4) **CFR-WASS** (Rosenbaum & Rubin, 1983)

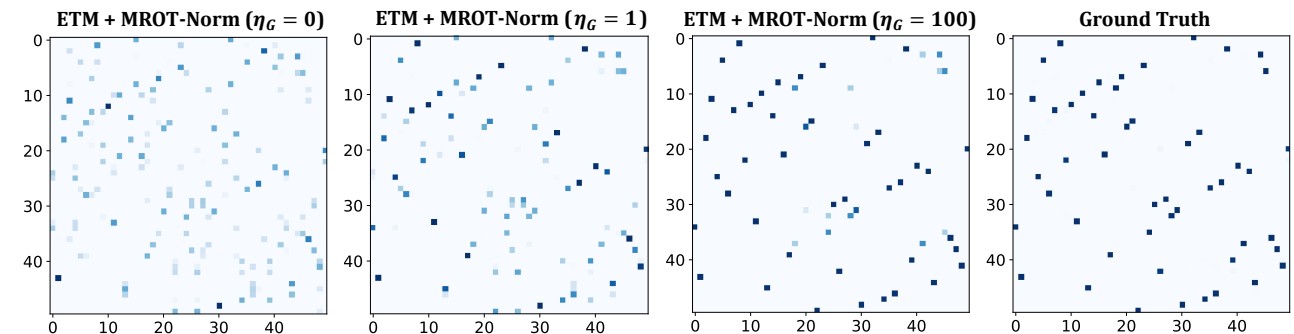

*Figure 6.* The matching results on ETM + MROT-Norm on SemiUOT with different values of $\eta_G = \{0, 1, 100\}$.

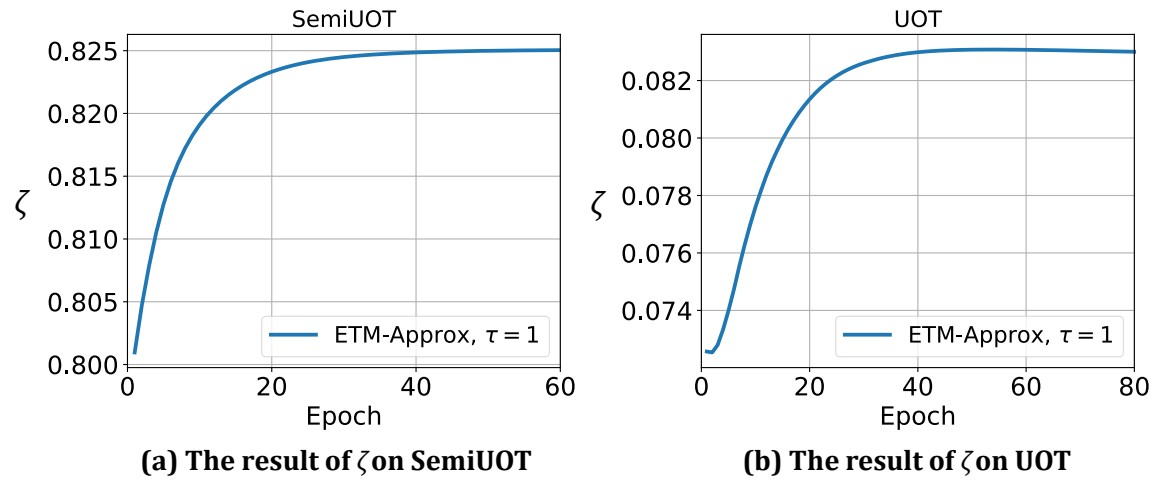

**(a) The result of $\zeta$ on SemiUOT**  **(b) The result of $\zeta$ on UOT**

*Figure 7.* The results of $\zeta$ and $L_P$ on UOT and SemiUOT.

utilizes standard optimal transport for adaptation, (5) **ESCFR** (Wang et al., 2023) further utilizes unbalanced optimal transport for adaptation. We adopt the same experimental settings as ESCFR (Wang et al., 2023). We utilize Precision in Estimation of Heterogeneous Effect (PEHE) (Shalit et al., 2017) and Area Under the Uplift Curve (AUUC) (Betlei et al., 2021) for the evaluation. Note that ESCFR + UOT(ETM + MROT) only replaces the entropic UOT in ESCFR with our proposed approximate-to-exact ETM +MROT. The experimental results are shown in Table 8. From that we can observe that ESCFR + UOT(ETM + MROT) achieves the best performance, indicating the efficacy of our proposed ETM method.

## L. More Experimental Results

**Parameter sensitivity.** We tune $\eta_G$ on SemiUOT via ETM-Refine with MROT-Norm in range of $\eta_G \in \{0, 1, 100\}$ using the same data samples shown in Fig.1 and show the results in Fig.6. We can observe that when $\eta_G$ is smaller (e.g., $\eta_G = 0$ or $\eta_G = 1$), the proposed KKT-multiplier regularization term $\mathcal{G}(\boldsymbol{\pi}, \boldsymbol{s}) = \langle \boldsymbol{\pi}, \boldsymbol{s} \rangle$ may struggle to play a significant role during the optimization process. Meanwhile when $\eta_G = 100$, ETM-Refine with MROT-Norm can achieve more accurate matching results comparing with the ground truth result. We can conclude that choosing larger value of $\eta_G$ can fully utilize the knowledge provided by KKT multiplier and enhance the final results. Therefore we set $\eta_G = 100$ empirically.

## M. Miscellaneous Discussions

**The role of $\zeta$ in ETM-based method.** We first discuss why we should involve translation invariant $\zeta$ in both $L_U$ and $L_P$. Specifically, we first analyse the case of SemiUOT. The Fenchel-Lagrange conjugate form of SemiUOT without translation invariant mechanism is given as:

$$\min_{\boldsymbol{f}, \boldsymbol{g}, \zeta}[\tau \sum_{i=1}^{M} a_i e^{-\frac{f_i}{\tau}} - \sum_{j=1}^{N} b_j g_j] \tag{59}$$

$$s.t. \ f_i + g_j \leq C_{ij}$$

We can adopt $c$-transform on Eq.(59) to obtain the unconstrained optimization problem as:

$$\min_{\boldsymbol{f}} \widetilde{L}_{\mathrm{P}} = \tau \sum_{i=1}^{M} a_i e^{-\frac{f_i}{\tau}} - \sum_{j=1}^{N} \inf_{k \in [M]} [C_{kj} - f_k] b_j, \tag{60}$$

We adopt L-BFGS to optimize $\widetilde{L}_{\mathrm{P}}$ using the same data samples as shown in Fig.1 with $\tau = 1$. Meanwhile, the translation invariant term $\zeta$ in SemiUOT should be calculated as follows:

$$\zeta = \tau \log \left( \sum_{i=1}^{M} a_i \exp \left( -\frac{f_i}{\tau} \right) \right) - \tau \log \left( \sum_{j=1}^{N} b_j \right) \tag{61}$$

Ideally, $\zeta$ should equals to 0 since $\sum_{i=1}^{M} a_i \exp \left( -\frac{f_i}{\tau} \right) = \sum_{j=1}^{N} b_j$. However, we can observe that $\zeta > 0$ during the iteration epoch on optimizing $\widetilde{L}_{\mathrm{P}}$ as shown in Fig.7(a). Therefore we can conclude that $\zeta$ is imdispenable during the calculation on SemiUOT. Likewise, the Fenchel-Lagrange conjugate form of UOT without translation invariant mechanism is given as:

$$\min_{\boldsymbol{v},\boldsymbol{u}} \left[ \tau_a \langle \boldsymbol{a}, e^{-\frac{\boldsymbol{u}}{\tau_a}} \rangle + \tau_b \langle \boldsymbol{b}, e^{-\frac{\boldsymbol{v}}{\tau_b}} \rangle \right] \tag{62}$$
$$s.t. \; u_i + v_j \leq C_{ij}.$$

Here we can adopt $c$-transform on Eq.(62) to obtain the unconstrained optimization problem as:

$$\min_{\boldsymbol{u}} \widetilde{L}_{\mathrm{U}} = \tau_a \sum_{i=1}^{M} a_i e^{-\frac{u_i}{\tau_a}} + \tau_b \sum_{j=1}^{N} b_j e^{\frac{\sup_{k=1}^{M} \left( u_k - C_{kj} \right)}{\tau_b}} \tag{63}$$

We also adopt L-BFGS to optimize $\widetilde{L}_{\mathrm{U}}$ using the same data samples as shown in Fig.2 with $\tau_a = \tau_b = 1$. Meanwhile, the translation invariant term $\zeta$ in UOT should be calculated as follows:

$$\zeta = \frac{\tau_a \tau_b}{\tau_a + \tau_b} \left[ \log \left\langle \boldsymbol{a}, \exp \left( -\frac{\boldsymbol{u}}{\tau_a} \right) \right\rangle - \log \left\langle \boldsymbol{b}, \exp \left( -\frac{\boldsymbol{v}}{\tau_b} \right) \right\rangle \right] \tag{64}$$

Ideally, $\zeta$ should equals to 0 since $\left\langle \boldsymbol{a}, \exp \left( -\frac{\boldsymbol{u}}{\tau_a} \right) \right\rangle = \left\langle \boldsymbol{b}, \exp \left( -\frac{\boldsymbol{v}}{\tau_b} \right) \right\rangle$. However, we can observe that $\zeta > 0$ during the iteration epoch on optimizing $\widetilde{L}_{\mathrm{U}}$ as shown in Fig.7(b). Therefore we can conclude that $\zeta$ is imdispenable during the calculation on UOT. In conclusion, the concept of translation invariant was first proposed in (Séjourné et al., 2022b). However, (Séjourné et al., 2022b) only utilizes translation invariant for entropic UOT. **We highlight that, in this paper, we further extend translation invariant for standard UOT/SemiUOT scenario**. We illustrate that translation invariant is essential in solving UOT and SemiUOT problems.