# OpenReview forum: "Unified (Semi) Unbalanced and Classic Optimal Transport with Equivalent Transformation Mechanism and KKT-Multiplier Regularization"
_ICML.cc/2025/Conference — Submitted to ICML 2025_

### Official Review · Reviewer_PKD1 · 2025-03-14

**Overall Recommendation:** 3

**Summary:**

This paper presents a new approach to the Semi-Unbalanced Optimal Transport (SemiUOT) problem by determining the marginal probability distribution using the Equivalent Transformation Mechanism and extends it to the Unbalanced Optimal Transport (UOT) problem. To improve matching accuracy, the authors introduce a KKT-Multiplier regularization term combined with the Multiplier Regularized Optimal Transport method. Experiments demonstrate the effectiveness of the proposed methods for UOT/SemiUOT problems.

## **update after rebuttal**
I appreciate the author's response and the additional experiments provided. Accurate computation should be the core aspect of the algorithm. While linear programming does involve a relatively high computational cost, this can be mitigated under specific conditions through hardware enhancements such as GPUs and multi-threading. Consequently, I will maintain the score I have assigned for now.

**Claims And Evidence:**

Yes, the claims made in the submission are supported by clear and convincing evidence.

**Essential References Not Discussed:**

The key contributing author is to improve the sinkhorn algorithm, but in the comparison experiments, it's all about using sinkhorn's baseline model for the comparison.

**Experimental Designs Or Analyses:**

Yes, I checked. The experimental design was comprehensive, covering both synthetic and real datasets, comparing multiple baseline methods, using reasonable assessment metrics, and validating the robustness of the method through parametric analysis and ablation experiments. However, a detailed description of parameter selection could further strengthen the credibility of the experiment.

**Methods And Evaluation Criteria:**

Yes, but the time complexity is not given in the convergence rate.

**Other Comments Or Suggestions:**

It is recommended that the authors compare their method with a variety of other computational OT methods beyond Sinkhorn. Additionally, the authors should consider using Total Variation as an alternative to KL divergence.

**Other Strengths And Weaknesses:**

The Equivalent Transformation Mechanism proposed in this paper surpasses existing optimal transport - based domain adaptive algorithms to some extent. However, it has the following limitations:
(1)	It fails to provide the convergence speed of the overall method.
(2)	Most compared models are based on Sinkhorn experiments. The authors don't compare them with other computational methods like linear programming - based experiments.
(3)	Since KL divergence can't handle non - overlapping distributions, it's questionable how the authors plan to use it in semiUOT for partial domain adaptation.

**Questions For Authors:**

It is recommended that the authors include other calculations of optimal transport in their comparison experiments, such as linear programming, approximation algorithms.

**Relation To Broader Scientific Literature:**

There is a correlation, but the authors have mainly improved the calculation of UOT, but not the algorithm.

**Theoretical Claims:**

Yes, I do. All basic theoretical derivations are correct.

---

> ### Author Rebuttal · Authors · 2025-03-29
>
> + Comment 1: The time complexity is not given in the convergence rate.
>
> Response 1: The computation complexity of ETM-Approx is $O(NM\log (1/\varepsilon_a))$ where $\varepsilon_a$ denotes the error tolerance (e.g., $ε_a = || \hat{f} - \hat{f}_o||_∞$ in SemiUOT and $ε_a = || \hat{u} - \hat{u}_o ||_∞$ in UOT, $\hat{f}_o$ and $\hat{u}_o$ denote the optimal solution on SemiUOT and UOT respectively). When the initial solution is sufficiently close to the optimal solution, quasi-Newton optimization procedure [1] can achieve super-linear convergence rate [1]. Thus, the time complexity for ETM-Refine is given as $O(NM\log (1/\varepsilon_a) + MN D_T)$ where $D_T$ denotes the number of iterations.
>
> Moreover, the convergence speed on MROT is determined by different kinds of regularization terms (e.g., entropy or norm regularization). For instance, ETM-Approx + MROT-Ent or ETM-Approx + MROT-Norm has the computation complexity as $O(NM\log (1/\varepsilon_a) + NMD_{\pi})$ where $D_{\pi}$ denotes the number of iteration. Likewise, ETM-Refine + MROT-Ent or ETM-Refine + MROT-Norm has the computation complexity as $O(NM\log (1/\varepsilon_a) + MN D_T + NMD_{\pi})$.
>
> + Comment 2: A detailed description of parameter selection should be provided.
>
> Response 2: We adopt the parameter sensitivity analysis on $\epsilon$ and report the results in Fig.4. Moreover, we provide more detailed results by varying $\epsilon = (0.01, 0.1, 1)$ and report the number of iterations with $\tau = (0.1,1)$ and report in **Response 3 for Reviewer JNNS (Table A)** due to the space limit.
>
> From this, we observe that a smaller value of $\epsilon=0.01$ provides a more accurate smoothness approximation [2], resulting in fewer iterations needed for refinement. Moreover, we also vary the hyper parameter of $\eta_G$ in Appendix L and report the results in Fig.6. We provide more detailed result by varying $\eta_G = (0, 1, 100)$ for SemiUOT/UOT with ETM + MROT-Entropy and ETM + MROT-Norm, where $\tau = 1$, $\eta_{\rm Reg} = 0.1$ and $N = 500$, and reported the absolute error $e = \sum_{i,j} || \pi_{ij} - \pi^*_{ij} ||_1 $ in **Response 3 for Reviewer JNNS (Table B)** due to the space limit.
>
> From that we can observe that a larger value on $\eta_G$ (e.g., $\eta_G = 100$) can provide more useful KKT-multiplier regularization and boost the model performance as also reflected in Fig. 6. Therefore we set $\epsilon=0.01$ to provide a more accurate smoothness function approximation and $\eta_G = 100$ to provide better KKT regularization in our experiments.
>
> + Comment 3: Other computational methods like linear programming should be reported.
>
> Response 3: Thank you for highlighting this important point. Following your advice, we present the detailed experimental results of the linear programming (LP) method combined with our proposed approach, as shown in the following table A.
>
> Table A. The time consumption (s) on synthetic data with $\tau = 1$
> | Method | $N$ = 500 | $N$ = 600 | $N$ = 700 | $N$ = 800| $N$ = 900 | $N$ = 1000 |
> | ------ | ------ | ------ | ------ | ------ | ------ | ------ |
> | UOT (ETM-Exact + MROT-Norm) | 5.43 | 8.33 | 11.07 | 15.76 | 21.96 | 33.99 |
> | UOT (ETM-Exact + LP) | 10.79 | 15.64 | 20.15 | 28.58 | 37.36 | 58.27 |
>
> We also conduct ETM-Exact + LP for domain adaptation tasks as shown in Table B.
>
> Table B. Classification accuracy on Office-Home for UDA
> | Method | Ar->Cl | Ar->Pr | Ar->Rw |  Cl->Ar | Cl->Pr |  Cl->Rw | Pr->Ar | Pr->Cl | Pr->Rw | Rw->Ar | Rw->Cl | Rw->Pr | Avg |
> | ------ | ------ | ------ | ------ | ------ | ------ | ------ | ------ | ------ | ------ | ------ | ------ | ------ | ------ |
> | JUMBOT + UOT(ETM + MROT-Ent) | 59.0 | 78.5 | 83.4 | 68.7 | 77.1 | 77.6 | 68.3 | 57.2 | 82.4 | 76.2 | 62.5 | 86.4 | 73.1 |
> | JUMBOT + UOT(ETM + MROT-Norm) | 59.4 | 78.7 | 84.1 | 68.5 | 77.3 | 78.5 | 68.6 | 57.9 | 82.8 | 76.3 | 62.5 | 86.5 | 73.4 |
> | JUMBOT + UOT(ETM + LP) | 60.7 | 79.2 | 84.8 | 68.7 | 77.9 | 79.0 | 69.1 | 58.3 | 83.1 | 76.6 | 62.7 | 87.2 | 73.9 |
>
> From that we can observe ETM + LP can provide more accurate results and boost the model performance. However, ETM-Exact + LP could be severely **time-consuming** compared with other methods (e.g., ETM-Exact + MROT-Norm).
>
> + Comment 4: It's questionable how the authors plan to use it in semiUOT for partial domain adaptation.
>
> Response 4: We adopt KL-divergence between the weights on data samples and the uniform distributions. SemiUOT can select and reweight the most similar data samples with higher values. 	We also conduct a toy example shown in Fig.1 where source and target domains are definitely non-overlapped. However, SemiUOT can still figure out outliers (the irrelevant data) and therefore it is reasonable to adopt SemiUOT to solve partial domain adaptation problems.
>
> **Reference**:
>
> [1] A quasi-Newton approach to non-smooth convex optimization
>
> [2] Smooth minimization of non-smooth functions.

---

### Official Review · Reviewer_JNNS · 2025-03-14

**Overall Recommendation:** 3

**Summary:**

The paper introduces a new method called the Equivalent Transformation Mechanism (ETM) that computes Unbalanced Optimal Transport (UOT) and Semi-Unbalanced Optimal Transport (SemiUOT) problems without relying on entropy regularization. The key idea is to compute the final marginal distributions explicitly through a dual optimization method then transform the UOT/SemiUOT problem into a classical OT problem without the relax on marginal constrain. This paper further proposed three variants of ETM for more efficient computation. Experiments were conducted on both synthetic data and domain adaptation tasks using proposed ETM variants thar are ETM-Exact, ETM-Approx, and ETM-Refine. The experiment results shows better performance compared to existing methods, particularly in terms of accuracy.

**Claims And Evidence:**

This paper majorly claims that ETM obtains exact marginal distributions for UOT and SemiUOT without the need for entropy regularization, leading to more accurate and sparse matching solutions. This claim is supported by the derivations showing how dual formulations and KKT conditions can yield the necessary marginal reweighting.

The other claim is about the efficiency of proposed ETM variants. While the empirical results supports this claim, the paper lacks an explicit theoretical analysis (e.g., Big-O complexity) of the proposed methods, leaving the claims about efficiency less supported.

**Essential References Not Discussed:**

I think the references are good.

**Experimental Designs Or Analyses:**

The paper provides comparisons between the proposed ETM variants and several state-of-the-art methods on both synthetic and domain adoptation task. The runtime and computation error analyses illustrate the trade-offs between different ETM variants. But, the experiment does not include the sensitive analysis on the hyperparameters.

**Methods And Evaluation Criteria:**

The evaluation on both synthetic datasets and domain adaptation tasks looks good to me for showing the usage of the method. But I think the evaluation on efficiency could be strengthened with more discussion on scalability, like how the method performs with larger datasets and a comparison regarding GPU acceleration.

**Other Comments Or Suggestions:**

na

**Other Strengths And Weaknesses:**

Strengths:

This paper introduces a new method that transforms UOT/SemiUOT problems into classical OT via explicit marginal reweighting. And the realated theoretical derivations are robust.

Weaknesses:

The implementation of ETM is not trivial due to its reliance on dual optimization method, L-BFGS. This framework is sequential iterative methods that limits the parallelizability and GPU acceleration.

Readibility, some concepts and notations such as SemiUOT and the variable $\tau$ are introduced late in the paper, making it difficult for readers unfamiliar with these terms to follow.

The paper lacks an theoretical time complexity analysis, mainly relying on empirical comparisons.

**Questions For Authors:**

Could you provide a formal Big-O complexity analysis for your proposed ETM algorithm variants, ETM-Approx and ETM-Refine?

Given the sequential nature of L-BFGS, do you see viable approaches for parallelizing or adapting your algorithm for GPU implementations?

Could you elaborate more on how sensitive your approach is to hyperparameters, such as $\tau$, $\epsilon$?

**Relation To Broader Scientific Literature:**

As they claimed, I think the key contributions of this paper is that it builds on recent work in UOT and SemiUOT, and proposed a method that avoids the drawbacks brought by entropy regularization.

**Theoretical Claims:**

I reviewed the main theoretical derivations, and the proofs look valid overall.

---

> ### Author Rebuttal · Authors · 2025-03-29
>
> + Comment 1: The time complexity is not provided.
>
> Response 1: The computation complexity of ETM-Approx is $O(NM\log (1/\varepsilon_a))$ where $\varepsilon_a$ denotes the error tolerance (e.g., $ε_a = || \hat{f} - \hat{f}_o||_∞$ in SemiUOT and $ε_a = || \hat{u} - \hat{u}_o ||_∞$ in UOT, $\hat{f}_o$ and $\hat{u}_o$ denote the optimal solution on SemiUOT and UOT respectively). When the initial solution is sufficiently close to the optimal solution, quasi-Newton optimization procedure [1] can achieve super-linear convergence rate [1]. Thus, the time complexity for ETM-Refine is given as $O(NM\log (1/ε_a) + MN D_T)$ where $D_T$ denotes the number of iterations.
>
> + Comment 2: How the method performs with larger datasets.
>
> Response 2: We solve the optimization problem on the GPU. We have conducted the experiments on the large transfer learning datasets, such as Office-Home. Specifically, Office-Home contains approximately 15,500 images, covering 65 categories, with images collected from office and home scenes. In our experiments, we set the batch size to 512. It takes approximately 3.7 seconds to perform one UOT computation, while one execution of ETM-Refine + MROT-Ent takes about 4.1 seconds.
>
> + Comment 3: Could you elaborate more on how sensitive your approach is to hyperparameters?
>
> Response 3: **We have conducted the parameter sensitivity in Section 5.3** by varying $\epsilon$ and reported the results in Fig.4. Moreover, **we also vary the hyper parameter of $\eta_G$ in Appendix L and report the results in Fig.6.** Based on your valuable comment, we collect $\epsilon = (0.01, 0.1, 1)$ and report the number of iterations  with $τ = (0.1,1)$ shown in the following Table A.
>
> Table A. Number of iterations to convergence on SemiUOT/UOT
> | Method | ETM-Refine ($\epsilon=0.01$) | ETM-Refine ($\epsilon=0.1$) | ETM-Refine ($\epsilon=1$)| ETM-Exact |
> | ------ | ------ | ------ | ------ |  ------ |
> | Number of iterations to convergence (SemiUOT $L_P$, $τ = 0.1$) | 135 | 189 | 215 | 243 |
> | Number of iterations to convergence (SemiUOT $L_P$, $τ = 1$) | 97 | 153 | 176 | 219 |
> | Number of iterations to convergence (UOT $L_U$, $τ = 0.1$) | 114 | 146 | 157 | 176 |
> | Number of iterations to convergence (UOT $L_U$, $τ = 1$) | 83 | 129 | 140 | 168 |
>
> Furthermore, we vary $η_G = (0, 1, 100)$ for SemiUOT/UOT with ETM + MROT-Entropy and ETM + MROT-Norm, where $τ = 1$, $\eta_{\rm Reg} = 0.1$ and $N = 500$, and reported the absolute error $e = \sum_{i,j} ||\pi_{ij} - \pi^*_{ij}||_1$ in the Table B:
>
> Table B. The absolute error on SemiUOT/UOT
> | Method | $η_G = 0$ | $η_G = 1$ | $η_G = 100$ |
> | ------ | ------ | ------ | ------ |
> | ETM + MROT-Entropy (SemiUOT) | 1.31  |  0.97  | 0.42  |
> | ETM + MROT-Norm (SemiUOT) |  0.79  | 0.64   |  0.28  |
> | ETM + MROT-Entropy (UOT) |   1.23  |  0.85  |  0.39  |
> | ETM + MROT-Norm (UOT) | 0.54  |  0.46  |  0.31  |
>
> Moreover we conduct the experiments with $\epsilon = (0.01, 0.1, 1)$ and $\eta_G = (0, 1, 100)$ on UDA in Office-Home with $τ = 1$ following JUMBOT (i.e., sensitivity analysis on $τ$ has been investigated in JUMBOT) and report the average classification below:
>
> |  | JUMBOT + UOT(ETM + MROT-Ent) | JUMBOT + UOT(ETM + MROT-Norm) |
> | ------ | ------ | ------ |
> | $\epsilon = 0.01, η_G = 100$ | 73.1 | 73.4 |
> | $\epsilon = 0.1, η_G = 100$ | 72.6 | 73.0 |
> | $\epsilon = 1, η_G = 100$ | 72.2 | 72.5 |
> | $\epsilon = 0.01, η_G = 1$ | 72.4 | 72.8 |
> | $\epsilon = 0.01, η_G = 0$ | 71.7 | 71.9 |
>
> From this, we observe that a smaller value of $\epsilon=0.01$ provides a more accurate smoothness approximation [4], resulting in fewer iterations needed for refinement. Meanwhile we can observe that a larger value on $η_G$ (e.g., $η_G = 100$) can provide more useful KKT-multiplier regularization and boost the model performance as also reflected in Fig. 6.
>
> + Comment 4: Given the sequential nature of L-BFGS, do you see viable approaches for parallelizing or adapting your algorithm for GPU implementations?
>
> Response 4: We leverage code from [1, 3] to optimize L-BFGS, though directly applying it to ETM-Exact may lack efficiency. Our key contribution in this paper is using fixed-point iteration in ETM-Approx to efficiently generate reliable initial solutions for ETM-Refine, achieving superlinear convergence as noted in [1, 4]. Parallelizing L-BFGS exceeds our current scope but is planned for future research.
>
> + Comment 5: Readability should be improved.
>
> Response 5: Thanks for your advice. We will first introduce the important notations at the beginning of our method in the final version to make the paper more readable.
>
> **Reference:**
>
> [1] A quasi-Newton approach to non-smooth convex optimization
>
> [2] Smooth minimization of non-smooth functions.
>
> [3] PyTorch: An Imperative Style, High-Performance Deep Learning Library
>
> [4] Optimization: Modeling, Algorithm and Theory

---

### Official Review · Reviewer_ewxK · 2025-03-14

**Overall Recommendation:** 3

**Summary:**

The paper proposes an approach of transforming the Unbalanced and Semi-unbalanced Optimal Transport (UOT/SUOT) problem into the classical OT problem. It is done by finding a scheme for proper reweighing of the marginal distributions. After this, the authors propose an approach for solving the discrete UOT/SUOT problems and test in a variety of experiments.

##  **After rebuttal.**
The authors have answered my questions. Thus, I update the score.

**Claims And Evidence:**

- The main claims made in the submission are supported by proofs and experiments. However, in lines 63-67, the authors write that the idea that "we can transform SUOT, UOT problems into classic OT problems by adjusting the weights" gives new insights on understanding of the UOT and SUOT problems. However, I can not agree with this point since the connection between these types of problems was already established in previous works, see (Choi et al., 2023, Theorem 3.3).

- In section 5.3 it is not clear in which experiment the authors conduct solvers comparison. It seems to be a synthetic experiment but it should be clearly written.

I also have some concerns regarding the experimental evaluation which I give below.

**References.**

Choi, J., Choi, J., & Kang, M. (2023). Generative modeling through the semi-dual formulation of unbalanced optimal transport. Advances in Neural Information Processing Systems, 36, 42433-42455.

**Essential References Not Discussed:**

The understanding of connection between the UOT/SUOT and classic OT problem was previously established in (Choi et al., 2023, Theorem 3.3). While this paper do not propose an algorithm to directly estimate the reweighted marginals for discrete measures, I think it is necessary to refer to this theoretical result.

The paper contributes to the field of discrete OT solvers. However, I think that the paper will benefit from stating the difference between discrete and continuous OT/UOT/SUOT solvers and referencing existing works in the field of continuous UOT. For example, the papers listed below are not included in the paper.

K. D. Yang and C. Uhler. Scalable unbalanced optimal transport using generative adversarial networks. In International Conference on Learning Representations, 2018.

F. Lübeck, C. Bunne, G. Gut, J.S. del Castillo, L.Pelkmans, and D.Alvarez-Melis.Neural unbalanced optimal transport via cycle-consistent semi-couplings. arXiv preprint arXiv:2209.15621, 2022.

M. Gazdieva, A. Asadulaev, E. Burnaev, and A. Korotin. Light Unbalanced Optimal Transport. Advances in Neural Information Processing Systems, 37, 2024.

L. Eyring, D. Klein, T. Uscidda, G. Palla, N. Kilbertus, Z. Akata, and F. Theis. Unbalancedness in neural monge maps improves unpaired domain translation. In The Twelfth International Conference on Learning Representations, 2024.

**Experimental Designs Or Analyses:**

I have concerns regarding the experimental designs:
- My major concern is devoted to the limited number of experimental setups considered in the paper. It seems hard to evaluate the quality of the proposed scheme for UOT/SUOT solution in the domain adaptation problem, since here the performance largely relies on many additional factors, e.g., the underlying approach for domain adaptation. I kindly suggest the authors to consider more synthetic examples. For example, synthetic experiments do not cover cases of datasets with **outliers** where UOT/SUOT-based approaches are usually used. It would be interesting to see how the approach performs in this experiment w.r.t. other approaches.
- I am also concerned by the overall performance of the proposed approach for the classic OT solution (MROT). The experiments which compare: 1) this MROT approach with other approaches for classic OT solutions, and 2) ETM approach for marginals reweighing plus MROT vs. ETM + other classic OT methods are missing.

**Methods And Evaluation Criteria:**

The method and evaluation criteria make sense.

**Other Comments Or Suggestions:**

N/A

**Other Strengths And Weaknesses:**

**Strengths.** The paper propose a direct algorithm for adjusting the weights in UOT/SUOT problem and, thus, converting it to the classic OT problem.

**Weaknesses.** The benefits of the proposed approach for classic OT estimation are not clear. The approach should be directly compared with other classic OT ones. See pervious sections for my other concerns.

**Questions For Authors:**

- As far as I understand, you propose a new approach (MROT) for finding the solutions of classic OT problem which, however, uses multipliers $s$ obtained from the reweighing step. I am wondering, is it possible to MROT for any classic OT problem when reweighing is not needed? If yes, then why you did not perform comparison with other approaches for classic OT?
- Why you did not perform quantitative comparison of your approach Ent-UOT (Pham et al., 2020) in section 5.3? It seems to be important to understand the performance of your approach.
- What is the computational complexity of your algorithms? It seems to be bigger than that of the Sinkhorn one.

**Relation To Broader Scientific Literature:**

The papers proposes a scheme of transforming the SUOT/UOT problem into the classic OT one by adjusting the weights of the marginals' points. It also proposes a new algorithm for classic OT problem solution.

**Theoretical Claims:**

I have skimmed through the theoretical claims.

---

> ### Author Rebuttal · Authors · 2025-03-28
>
> + Comment 1:  The differences/novelty between this paper and Theorem 3.3 in [Choi] should be highlighted.
>
> Response 1: Theorem 3.3 in [Choi] and our proposed ETM differ significantly in several aspects:
>
> (1) Theorem 3.3 mainly considers the continuous case and does not involve the translation invariant term $ζ$. As we shown in Appendix M, without $ζ$, the transformed marginal probability will not be equal in the practice and therefore [Choi] cannot guarantee SemiUOT/UOT can be transformed into classic OT, making it impractical in the discrete scenario.
>
> (2) [Choi] primarily discusses UOT and does not explore SemiUOT. Our proposed ETM method specifically addresses the discrete case by directly calculating the exact value on dual Lagrange multipliers (e.g., $f$ and $u$ in SemiUOT/UOT) for data reweighting and finding $π$, a topic not covered by [Choi].
>
> In summary, our proposed ETM method ensures that the transformed marginal probability is equal in the discrete scenario, making the equivalent transformation **practical** for further obtaining $π$ with KKT regularization.
>
> + Comment 2: Descriptions in section 5.3 are not clear.
>
> Response 2: We conduct solver comparison on synthetic datasets in Fig.3. That is, we sample 90\% data from $P_X$ and $P_Z$ accordingly while also randomly sampling 10\% outlier data for $P_{X}$ and $P_{Z}$ to create synthetic dataset and conduct the experiments w.r.t. other methods.
>
> + Comment 3: The domain adaptation results may rely on additional factors.
>
> Response 3: We follow the same framework, loss functions and experimental settings as the famous models JUMBOT/MOT and only replace UOT/SemiUOT solver with ETM as detailed in line 372-379. That is, we control the variants of these additional factors in the experiments.
>
> + Comment 4: ETM+MROT vs. ETM+other classic OT methods are missing.
>
> Response 4: The results for ETM combined with Entropy and Norm under SemiUOT/UOT scenarios, using 500 synthetic data, are presented below:
>
> | Method | $τ$=0.01 | $τ$=0.1 | $τ$=1| $τ$=10 | $τ$=100|
> | ------ | ------ | ------ | ------ | ------ | ------ |
> | (SemiUOT) ETM+Entropy | 0.15 | 0.73 | 1.31 | 1.74 | 1.89 |
> | (SemiUOT) ETM+Norm | 0.11 | 0.48 | 0.79 | 0.96 | 1.24 |
> | (UOT) ETM+Entropy | 0.10 | 0.61 | 1.23 | 1.57 | 1.78 |
> | (UOT) ETM+Norm | 0.08 | 0.35 | 0.54 | 0.71 | 0.96 |
>
> Both ETM+Entropy and ETM+Norm performs poorly when $η_{G}=0$ comparing with the results in Fig.3(c)-(d) due to insufficient guidance from the KKT conditions. It also reflects the visualization results in Fig.6 in our paper.
>
> + Comment 5:  Some similar related work should be added.
>
> Response 5: We will add these references into our main paper with discussions.
>
> + Comment 6: Is it possible to MROT for any classic OT problem when reweighing is not needed?
>
> Response 6: No, MROT requires multipliers $s$ via the ETM method for data sample reweighting, tailored for SemiUOT/UOT. The multipliers $s$ cannot be derived without the reweighing step, rendering MROT incapable of addressing classic OT problems. Determining $s$ for classic OT problems presents its own challenges and falls outside the scope of this paper. Our paper primarily focuses on solving $\pi$ for SemiUOT/UOT, and we have provided ample experiments to validate our proposed KKT-multiplier regularization term, which yields more precise matching results.
>
> + Comment 7: The performance of Ent-UOT should be provided.
>
> Response 7: We provide the experimental results (i.e., time consumption and absolute error) of Ent-UOT in Table A and Table B respectively.
>
> Table A: The time consumption of Ent-UOT where $\tau = 1$ with synthetic data.
> | Method | $N$ = 100 | $N$ = 200 | $N$ = 300| $N$ = 400 | $N$ = 500| $N$ = 600 | $N$ = 700 | $N$ = 800| $N$ = 900 | $N$ = 1000|
> | ------ | ------ | ------ | ------ | ------ | ------ | ------ | ------ | ------ | ------ | ------ |
> | ETM-Refine + MROT-Ent | 0.13 | 0.38| 0.79 | 1.48 | 2.98 | 5.03 | 6.60 | 9.75 | 11.93 | 19.98 |
> | ETM-Approx + MROT-Ent | 0.12 |  0.32  |  0.58  |  1.13  |  2.03 |  3.01 |  4.94 |  7.12 | 10.89  | 17.45  |
> | Ent-UOT | 0.11 | 0.28 | 0.55 | 1.08 | 2.15| 3.02 | 5.84 | 7.68 | 10.45 | 17.62 |
>
> Table B: The absolute error of Ent-UOT with synthetic data.
> | Method | $τ$=0.01 | $τ$=0.1 | $τ$=1| $τ$=10 | $τ$=100|
> | ------ | ------ | ------ | ------ | ------ | ------ |
> | ETM-Approx + MROT-Ent | 0.05 | 0.19 | 0.31 | 0.50 | 0.54 |
> | Ent-UOT | 0.12 | 0.64 | 1.25 | 1.61 | 1.83 |
>
> We can observe that Ent-UOT could lead to coarse output matching results and it further illustrates the efficacy of our proposed ETM method.
>
> + Comment 8: What is the computational complexity of your algorithms?
>
> Response 8: The computation complexity of ETM-Approx/ETM-Refine with MROT-Ent is $O(NM)$ which has the same BigO magnitude as the Sinkhorn algorithm. Empirically, ETM-Approx with MROT-Ent and Ent-UOT (Sinkhorn in UOT) have roughly comparable running times while less absolute error as reported in Response 7 Table A-B.

---

> > ### Comment · Reviewer_ewxK · 2025-04-03
> >
> > Thank you for your answers, you have mitigated most of my concerns. I incorrectly pointed out to the experiment with outliers which is indeed included in the paper. But did you consider synthetic experiment with imbalance of classes in the source and target measures (e.g., in the context of Gaussian mixtures)? The property of dealing with class imbalance is an another nice feature of UOT-based approaches. It seems to be a valid additional experiment justifying the properties of your method.

---

> > > ### Author Response · Authors · 2025-04-04
> > >
> > > Esteemed Reviewer,
> > >
> > > Thank you for your kind message, and valuable comments helping us improve and refine our manuscript.
> > >
> > > + Comment A: Did you consider some synthetic experiments with imbalance of classes in the source and target measures (e.g., in the context of Gaussian mixtures)?
> > >
> > > Response A: **Sure, our proposed ETM-based method can tackle the class imbalance scenario.** Specifically, we conduct the experiments following the settings in [1] (shown in Fig.2 in [1]) where the source and target data distributions (Gaussian mixtures of two uniform distributions with different weights) are defined as $P_{{X}} = 2/5 U([-1,1] \times [0.5,1.5]) + 3/5U([5,6] \times [0.5,1.5])$ and $P_{{Z}} = 3/5 U([-1,1] \times [-0.5,-1.5])+ 2/5 U([5,6] \times [-0.5,-1.5])$, respectively. We first sample $N = 50$ data for both $P_{{X}}$ and $P_{{Z}}$ with $\tau = 0.1$ or $\tau = 0.9$ and conduct the UOT matching experiments accordingly. *(Note that No.1-No.20 data in $P_X$ are sampled from $U([-1,1] \times [0.5,1.5])$ and No.21-No.50 data in $P_X$ are sampled from $U([5,6] \times [0.5,1.5])$. Meanwhile No.1-No.30 data in $P_Z$ are sampled from $U([-1,1] \times [-0.5,-1.5])$ and No.31-No.50 data in $P_Z$ are sampled from $U([5,6] \times [-0.5,-1.5])$ to setup the synthetic data experiment).* The results can be found in the following anonymous link: https://anonymous.4open.science/r/ETM_matching/icml4808_tau_01_match.pdf and https://anonymous.4open.science/r/ETM_matching/icml4808_tau_09_match.pdf.
> > >
> > > Here the blue ‘+’ and red ‘x’ denote the source and target samples, respectively. The data distributions are set to be **class-imbalanced.** From that we can observe: (1) Ent-UOT could only provide coarse and inaccurate matching results. Moreover, Ent-UOT may lead to mismatch when $\tau = 0.9$. (2) GEMUOT with square norm regularization term can obtain more sparse matching results. However, the output results of GEMUOT are still far away from the ground truth. (3) **Our proposed ETM+MROT-Norm can achieve more accurate results meanwhile avoiding mismatch compared with Ent-UOT and GEMUOT, which indicates the efficacy of the proposed method.**
> > >
> > > Moreover, we collect the absolute error $e = \sum_{i,j} ||\pi_{ij} - \pi^*_{ij}||_1$ with 500 data samples on UOT as shown in the following table.
> > >
> > > | Method  | $\tau=0.1$, $N = 500$ | $\tau=0.9$, $N = 500$ |
> > > | ------ | ------ | ------ |
> > > | Ent-UOT | 0.59 | 1.12  |
> > > | GEMUOT | 0.38 | 0.60 |
> > > | ETM-Approx + MROT-Ent | 0.21 | 0.33 |
> > > | ETM-Refine + MROT-Ent  | 0.19 | 0.32 |
> > > | ETM-Refine + MROT-Norm  | 0.15 | 0.27 |
> > >
> > > **We also observe that our proposed ETM-based method can obtain more accurate output results for tackling the class imbalance scenario.** We will add these contents into the final version of our paper.
> > >
> > > *We hope that we addressed mainly of your concerns sufficiently, and if you agree, we would kindly request for updating the review in light of this response. If there is anything else we can answer or explain or discuss further, kindly do let us know.*
> > >
> > > Kind regards,
> > >
> > > Authors
> > >
> > > **Reference:**
> > >
> > > [1] Eyring L, Klein D, Uscidda T, et al. Unbalancedness in Neural Monge Maps Improves Unpaired Domain Translation. The Twelfth International Conference on Learning Representations.

---

### Decision · Program_Chairs · 2025-05-01

**Decision:**

Reject

**Comment:**

The paper proposes casting the semi-UOT problem as classical OT. This is done in multiple ways, exact, approximate etc., leading to different results. Optimization algorithms for solving the variants are discussed. Simulations show the methodology's efficacy.

The effect of the lse based approximation is not analysed and left as a hack. Analysis of the quality of this approximation will help the presentation. Some issues with simulations, presentation are pointed out by the reviewers. A computational analysis will help the presentation. All the reviews are borderline with no strong support for accepting the paper.